# Performance of long-chain alkane–1,mid–chain–diol based temperature and productivity proxies at test: a five-years sediment trap record from the Mauritanian upwelling

Gerard J. M. Versteegh [1,2,3], Karin A. F., Zonneveld[3,4], Jens Hefter[1,3], Oscar E. Romero[3], Gerhard Fischer[4] Gesine Mollenhauer[1,3]

[1]Alfred–Wegener-Institute for Polar and Marine Research, Am Handelshafen 12, D-27570 Bremerhaven, Germany

[2]Department of Physics and Earth Sciences, Jakobs University, Bremen, Bremen, 28759, Germany

[3]MARUM - Zentrum für Marine Umweltwissenschaften, Universität Bremen, Bremen, 28359, Germany

[4]Fachbereich Geowissenschaften, Universität Bremen, Bremen, 28359, Germany

*Correspondence to*: Gerard J. M. Versteegh (gerard.versteegh@awi.de)

**Abstract.** Long-chain alkane–1,mid–chain–diol (shortly diol) based proxies obtain increasing interest to reconstruct past upper ocean temperature and productivity. Here we evaluate performance of the sea

surface temperature proxies; long chain diol index (LDI), the diol saturation index (DSI) and the diol chain-length index (DCI), productivity/upwelling intensity proxies: two diol indices $DI_R$ and $DI_W$ and the combined diol index (CDI), as well as the nutrient diol index (NDI) as proxy for phosphate and nitrate levels. This evaluation is based on comparison of the diols in sediment trap samples from the upwelling region off NW Africa collected at 1.28 km water depth with daily satellite derived sea surface

temperatures (SST), subsurface temperatures, productivity, the plankton composition from the trap location, monthly phosphate and nitrate concentrations, wind speed and wind direction from the nearby Nouadhibou airport. The diol based SST reconstructions are also compared the long chain alkenones based SST reconstructions.

The alkenone SST correlate best with satellite SST ($r^2 = 0.60$). Amplitude and absolute values agree very well as do the flux corrected time series averages. For the diol proxies the situation is more complicated.

Diol proxies including 1,14 diols lag trade wind speed by 30 days. Since wind is nearly always from the NNE to NNW and induces the upwelling, we relate the variability in these proxies to upwelling induced processes. Correlation with the abundance of upwelling species and wind speed is best for the NDI and the 1,14 diol–based DCI and DSI. The $DI_R$, $DI_W$ and CDI perform comparatively poorly. A negative correlation between DSI and wind speed, may suggest that the DSI reflects wind speed forced upwelling related reductions temperature rather than irradiation induced temperatures. The nutrient proxy NDI shows no significant correlation to monthly phosphate and nitrate concentrations in the upper waters and a negative correlation with both wind-induced upwelling ($r^2=0.28$ and lagging 32 days) and the abundance of upwelling species ($r^2=0.38$). It is suggested that this proxy reflects upwelling intensity rather than upper ocean nutrient concentrations.

At the trap site, satellite SST lags wind speed forced upwelling by about 4 months. The 1,13 and 1,15 diol–based LDI derived SST lag satellite SST by 41 days but correlate poorly ($r^2 = 0.17$). Absolute as well as flux corrected LDI SST are on average 3°C too high and rather reflect values prevailing during the more oligotrophic summer period. Outliers to low LDI SST we attribute to 1,13 diols added during short, upwelling related, events. The use of the LDI in regions with higher productivity is therefore not recommended.

It appears thus that at the trap site the 1,14 diols primarily reflect conditions relating to upwelling whereas the $1,15C_{30}$ and to a lesser extent the 1,13 diols seem to reflect the conditions of the more oligotrophic ocean.

# 1 Introduction

Upper ocean temperature and productivity reconstructions are important for assessing past climate and environment. For organisms, optimal functioning of their membranes and transport in the cell is crucial and since temperature has a large influence on the solubility and viscosity of lipids, organisms adapt their lipid composition to temperature. Common responses to a decrease in temperature are (1) reducing

average chain-length of the lipid molecules, (2) increasing the number of double bonds or (3) the degree of cyclisation (e.g. Suutari et al., 1994; Elling et al., 2015; Sollich et al., 2017). This response of organisms to adapt their metabolism and metabolite composition to prevailing ocean conditions we can use to reconstruct past ocean temperature and productivity through searching for relevant metabolites in

the fossil record and taking them as environmental proxies. This is not an unequivocal enterprise since the metabolite composition of organisms is often influenced by a combination of environmental variables and, therefore, proxies based on these metabolites bear the risk of reflecting this. Furthermore, the same metabolites may be produced by different organisms, each having its own complex response to environment. To obtain robust and reliable temperature, nutrient and productivity reconstructions from

fossil metabolites it is essential to test the performance of these proxies in present day conditions. In this study, we do so by investigating the long-chain mid-chain diol composition in sediment trap samples in relation to environment during a multiyear trap deployment. We performed this study in the upwelling area off Cape Blanc, one of the most productive regions in the world (Chavez and Messié, 2009). From this region detailed (daily to monthly) records of water temperature, productivity, nutrient

concentrations and upwelling dynamics in the upper ocean as well as atmospheric parameters like wind direction and wind intensity are available.

Our study focuses on several diol-based proxies that have been proposed for temperature, productivity and/or upper ocean nutrient concentrations. For temperature these are (1) the Long-chain Diol Index (LDI) (Rampen et al., 2012), (2) the Diol Saturation Index (DSI) (Rampen et al., 2014a) and (3) the

Diol Chain length Index (DCI) (Rampen et al., 2009, 2014a). The Combined Diol Index (CDI) (Rampen et al., 2014a) and two Diol Indices (DI), the $DI_R$ (Rampen et al., 2008) and $DI_W$ (Willmott et al., 2010) have been suggested as proxies for upper ocean upwelling/productivity. The Nutrient Diol Index (NDI) has been proposed for reconstructing sea surface nitrate/phosphate concentrations (Gal et al., 2018).

Other diol-based proxies have been defined but these either strongly correlate with the proxies

mentioned above or are not relevant in the Mauritanian setting such as a diol proxy for terrestrial/fresh water input (Versteegh et al., 1997; Lattaud et al., 2017a,b). We also evaluate the diol temperature proxies in relation to the lipid based temperature proxy: the long-chain alkenone-based $U^{K'}_{37}$ (Prahl and Wakeham 1987; Brassell et al., 1986a).

## 1.1 Regional oceanography (as Heading 2)

The Mauritania upwelling system is part of the Canary Current (CC) Eastern Boundary Upwelling Ecosystem (CC–EBUE). The coastal upwelling off Mauretania is driven by the NNW to NNE trade winds and occurs where the southward CC flowing along the coast meets the northward Cap Verde Current (CVC) and Poleward Undercurrent (PUC). These currents are deflected offshore, resulting in the SW directed Cape Verde Frontal Zone (CVFZ) (Fig. 1). The result of this deflection and the offshore water export is visible as the giant Mauritanian chlorophyll filament (Fig. 1 after Romero et al. 2020). As a result of the coastal, shelf and slope topography and the ocean currents and trade winds from the North, the coastal region off Mauritania is characterized by almost permanent upwelling. Its intensity varies throughout the year with maximal intensity and extension in boreal winter and spring (Lathuilière et al., 2008; Cropper et al., 2014; Romero et al., 2020; Fig. 2). The offshore transport by the upwelling filaments is considerable and it has been estimated that during periods of intense upwelling 80% of the shelf particular matter production is transported into the open ocean up to 400 km offshore whereas remineralisation and turbulence sustain production and enhanced transport of organic carbon to 2000 km westwards (e.g. Gabric et al., 1993; Lovecchio et al., 2017, 2018).

## 2 Material and Methods

### 2.1 Sample collection

Particulate organic matter forming the base of this study was collected between June 2003 and March 2008 off Cape Blanc at the eutrophic mooring station CBeu (Fig. 1; Table 1). For the trap type and sampling performance see Romero et al. (2020). Classical cone-shaped traps with a surface opening area of 0.5 m$^2$ were used (Kiel SMT 230/234, Kremling et al., 1998). Over the five years studied, 120 samples were collected in total. Prior to each deployment, sampling cups were poisoned with 1mL of concentrated $HgCl_2$ per 100mL of filtered seawater. Pure NaCl was used to increase the density in the sampling cups up to 40 ‰. Upon recovery, samples were stored at 4 °C and wet split in the MARUM sediment trap laboratory at Bremen University, Bremen, using a rotating McLane wet splitter system. The largest swimmers, such as crustaceans, were handpicked with forceps and remaining swimmers > 1

mm were removed by carefully filtering through a 1 mm sieve prior to splitting. All flux data hereafter refer to the size fraction of < 1 mm.

## 2.2 Lipid extraction

Lipids were extracted using ultrasonic disruptor probes with successively less polar solvents: MeOH, MeOH–DCM (1:1, v–v) and DCM (after Müller et al., 1998). The TLEs were dried over anhydrous

$Na_2SO_4$ and saponified with 0.1 M KOH in 90/10 $MeOH:H_2O$ for 2 hours at 80°C. For desalting, each sample was washed with distilled water and a solution of DCM:MeOH (1:1, vol:vol). The supernatant was removed, the DCM:MeOH phases were rotary evaporated and later dried over anhydrous $Na_2SO_4$. Total lipid extracts were saponified with 6% KOH in MeOH at 85°C for 2 h. The neutral lipid fraction was extracted with hexane, and then fractionated into three polarity fractions using Bond–Elut silica gel

cartridges. Fractions 1–3 were eluted in 2mL each of hexane, hexane:dichloromethane 1:2 v–v, and methanol, respectively. For the samples from the 5[th] deployment of the eutrophic Cap Blanc trap (CBeu5) androstanol was used as internal standard.

## 2.3 Lipid data acquisition

For diol analyses the polar fractions were silylated with 100 µl N,O–bis (trimethylsilyl)

trifluoroacetamide (BSTFA) for 1 hour at 70°C. Except CBeu5 (see below), the concentrations of the different diol isomers were determined using a time of flight mass spectrometer (TOF–MS) LECO Pegasus III (LECO Corp., St. Joseph, MI) interfaced to an Agilent 6890 gas chromatograph (GC). The GC was equipped with a 15m x 0.18mm i.d. Rtx–1MS (Restek Corp., USA) column (film thickness: 0.10µm) with an integrated 5m guard column. A temperature-programmable cooled injection system

(CIS4, Gerstel) in combination with an automated liner exchange facility (ALEX, Gerstel) and a multi-purpose autosampler (MPS 2, Gerstel) were used for sample injection. The CIS was operated in solvent vent mode (2.5 µL injection volume) and initially held at 40 °C for 0.05 min, then heated at 12 °C $s^{-1}$ to 80 °C (held 0.1 min) with the split valve open, a vent flow of 100 mL $min^{-1}$ and a vent time of 10 sec. After the solvent vent time, the split valve was closed, the CIS heated at 12 °C $s^{-1}$ to 340 °C and held for

2 min for sample transfer to the GC-column. The GC oven was initially held at 60 °C for 1 min, then

heated at 50 °C min$^{-1}$ to 250 °C and at 30 °C min$^{-1}$ to 310 °C (held 2.5 min), resulting in an analysis time of 9.3 min per sample. Helium was used as carrier gas in constant flow mode at a rate of 1.5 mL min$^{-1}$.

The ion source was operated at a temperature of 220 °C with the GC connected to the source by means of a heated transfer line set to 280 °C. The pressure in the flight tube of the MS was 2 x 10$^{-7}$ Torr. Full range mass spectra from $m/z$ 50–600 obtained under EI conditions at 70 eV were recorded at a data rate of 40 spectra s$^{-1}$. Processing of the TOF–MS data was accomplished with the software (ChromaTOF™) provided by the manufacturer and included automated baseline and peak finding, spectral deconvolution of overlapping chromatographic peaks, peak area calculations, and extraction of compound specific $m/z$ (mass-to-charge ratio) ion chromatograms.

Relative amounts of alkyl diol isomers were estimated from peak areas of specific ions resulting from α–cleavage of trimethylsilyl-ethers at various mid-chain positions (de Leeuw et al., 1981) As an example, for $C_{28}$ diols peaks at $m/z$ 299 (1,14-diol) and $m/z$ 313 (1,13-diol), and for $C_{30}$ diols peaks at $m/z$ 313 (1,15 diol) and $m/z$ 327 (1,14-diols) were integrated (Rampen et al., 2008; Versteegh et al., 1997). For the diol quantification, selected reference samples containing alkyl diols in high relative amounts and without other interfering compounds (as indicated by GC/TOF–MS analyses) were additionally analyzed by GC–FID on an Agilent 6890 GC equipped with a 60m x 0.32mm i.d. DB1–MS (Agilent J&W) column (film thickness: 0.25μm). Samples were injected via an on-column injector. The GC oven was initially held at 60 °C for 3 min, and then heated at 20 °C min$^{-1}$ to 150 °C and at 6 °C min$^{-1}$ to 320 °C (held 30 min). Helium was used as carrier gas in constant flow mode at a rate of 1 mL min$^{-1}$. Alkyl diols were identified by retention times, elution order and relative abundances in comparison to the corresponding GC/TOF–MS analysis from which MS response factors for each diol were obtained. GC–FID areas of the respective diols were used for quantification by comparison to the peak area of an external $n$-$C_{28}$ 1-alkanol standard. If the diol peaks consisted of a coeluting isomeric mixture, isomer proportions were quantified by their relative ratios as obtained by analysis of isomer-specific $m/z$ ions from GC/TOF–MS. Quantified amounts of diols were then used to obtain mass-specific response factors, that allowed subsequent direct quantification of diols by GC/TOF–MS for the rest of the samples.

The diols of CBeu 5 were analyzed by GC–MS using an Agilent 6850 GC coupled to an Agilent 5975C MSD equipped with a fused silica capillary column (Restek Rxi-1ms; length 30 m; diameter 250 µm; film thickness 0.25 µm). The temperature program for the oven was as follows: held at 60 °C for 3 min, increased to 150 °C at 20 °C min$^{-1}$, increased to 320 °C at 4 °C min$^{-1}$, held at 320 °C for 15 min. Helium was used as carrier gas and the flow was held constant at 1.2mL min$^{-1}$. The MS source was held at 230 °C and the quadrupole at 150 °C. The electron impact ionization energy of the source was 70 eV.

Relative amounts of alkyl diol isomers were estimated from peak areas of specific ions as described above. Absolute amounts of diols were obtained by comparison to the peak area of the known amount of the internal standard androstanol (*m/z* 333).

## 2.4 Calculation of diol indices

Diol indices were calculated on the basis of relative abundances of the diol isomers.

### 2.4.1 Temperature proxies

Long Chain Diol Index (Rampen et al., 2012).

$$LDI = 1,15C_{30} / (1,13C_{28} + 1,13C_{30} + 1,15C_{30}) \tag{1}$$

Sea surface temperatures have been calculated based on the following transfer functions:

$$SST = (LDI — 0.095) / 0.033 = 30.3\ LDI — 2.88\ \text{(Rampen et al., 2012)} \tag{2}$$

$$SST = (LDI — 0.1082) / 0.0325 = 30.8\ LDI — 3.33\ \text{(DeBar et al., 2020)} \tag{3}$$

For this study the differences between both calibrations are negligible (< 0.1°C). The calibration Rampen et al. (2012) was used.

Diol Saturation Index (Rampen et al., 2014a)

$$DSI = (1,14C_{28} + 1,14C_{30} ) / (1,14C_{28} + 1,14C_{28:1} + 1,14C_{30} + 1,14C_{30:1}). \tag{4}$$

According to Rampen et al. (2014a) this index has a high correlation to temperature in *Proboscia* cultures (Rampen et al., 2009) following the relation:

$$T_{culture} = 40\ DSI + 1.2 \tag{5}$$

Diol Chain length Index (Rampen et al., 2009, 2014a)

$$DCI = 1,14C_{30} / (1,14C_{28} + 1,14C_{30}) \tag{6}$$

transfer functions have been published for this index on the basis of *Proboscia* diatom cultures:

$$T_{culture} = 50*DCI + 3.95 \tag{7}$$

on the basis of SE Atlantic core top values in relation to summer (February) SST (Rampen et al., 2009):

$$SST_{February} = 19.6*DCI + 14.0 \ (r^2 = 0.62, p = 3.2 \ 10^{-6}) \tag{8}$$

and calculated on the basis of the SE Atlantic data of Versteegh et al. (2000):

$$SST_{December} = 22.8*DCI + 6.34 \ (r^2 = 0.68, p = 1.5 \ 10^{-4}) \tag{9}$$

The DSI and DCI showed high correlations to culture temperature but these correlations were absent in
a global survey relating DSI and DCI derived from core-tops to SST (Rampen et al., 2014a).

## 2.4.2 Upwelling / upper ocean productivity proxies

Diol Index (Rampen et al., 2008)

$$DI_R = (1,14C_{28} + 1,14C_{30}) / (1,14C_{28} + 1,14C_{30} + 1,15C_{30}) \tag{10}$$

This has been suggested as a proxy for southwest monsoon upwelling in the Arabian Sea (Rampen et
al., 2008) and is also applicable to the Namibian upwelling (see Versteegh et al., 2000)

Diol Index (Willmott et al., 2010)

$$DI_W = (1,14C_{28} + 1,14C_{30}) / (1,14C_{28} + 1,14C_{30} + 1,13C_{28} + 1,13C_{30}) \tag{11}$$

This ratio was designed as a measure of the contribution of *Proboscia* diols relative to other diols.

Combined Diol Index (Rampen et al., 2014a)

$$CDI = (1,14C_{28} + 1,14C_{30}) / (1,13C_{28} + 1,13C_{30} + 1,14C_{28} + 1,14C_{30} + 1,15C_{30}) \tag{12}$$

## 2.4.3 Upper ocean nitrate and phosphate concentration proxies

Nutrient Diol Index (NDI) (Gal et al., 2018)

$NDI = (1,14C_{28} + 1,14C_{28:1}) / (1,14C_{28} + 1,14C_{28:1} + 1,14C_{30} + 1,14C_{30:1} + 1,13C_{28} + 1,13C_{30} + 1,15C_{30})$

(13)

$[PO_4^{3-}] = (NDI - 0.015) / 0.413 = 2.42\ NDI - 0.0363$ (14)

$[NO_3^-] = (NDI - 0.075) / 0.026 = 38.5\ NDI - 2.88$ (15)

As may be expected, the ratio of the slopes of these transfer functions is 1:16, the Redfield Ratio
(Redfield 1963).

Generally, fluxes recorded in sediment traps have a logarithmic nature. As a result, linear correlations between diols are subject to bias by single high values. To overcome this bias, we based our correlations on the log-transformed diol concentrations (Sup. Fig. S1). Diol concentrations of CBeu1–4 are on
average more than one order of magnitude (20.71 times) lower than for CBeu5 (Sup. Fig. S2). Although this does not affect the diol proxy ratios, it does affect the direct comparison of diol fluxes. Concentrations of CBeu1–4 have been calculated with an external standard and by using the peak areas, response factors, injection volume and sediment mass. In contrast, diol concentrations of CBeu5 have been calculated using peak areas, response factors and an internal standard and these fluxes are in good
agreement with those reported from other sediment traps from high productivity regions (Rampen et al., 2008, DeBar et al., 2019). For comparison of diol concentrations between both trap series we multiplied the CBeu1–4 values by 20.71 to agree with those of the better–validated CBeu5 concentrations. Obviously, these new values do not represent exact concentrations but they reduce the concentration error considerably whereas the concentration dynamics between the individual samples remain intact
(Sup. Fig. S2).

## 2.5 Alkenone based $U^{K'}_{37}$ SST proxy

Lipid SST proxy data on long chain alkenone based $U^{K'}_{37}$ have been determined for trap CBeu5 (material collected between 28 March 2007 and 17 March 2008). Data have been combined with those of Mollenhauer et al. (2015) for CBeu1–4.
The alkenones in fraction 2 were analysed using an Agilent 5890 gas chromatograph equipped with a DB5–MS capillary column and a flame ionization detector. Alkenone identification is by relative

retention times and comparison with a laboratory-internal standard sediment. The $U^{K'}_{37}$ index was calculated using the peak areas of the $C_{37:2}$ and $C_{37:3}$ alkenones (Prahl and Wakeham, 1987):

$$U^{K'}_{37} = [C_{37:2}] / ([C_{37:2}] + [C_{37:3}]) \tag{16}$$

Analytical precision based on repeated analyses of the standard sediment is ± 0.01 units of the $U^{K'}_{37}$ index. The $U^{K'}_{37}$ values were converted to temperatures using the calibration for suspended particulate matter (Conte et al., 2006):


$$SST_{UK} (°C) = -0.957 + 54.293 \, U^{K'}_{37} - 52.894(U^{K'}_{37})^2 + 28.321 \, (U^{K'}_{37})^3 \tag{17}$$

The average $SST_{UK}$ of CBeu1–4 (Mollenhauer et al., 2015) is lower than that of CBeu5 (t-test for the same mean p = 0.0014; F-test for same variance p= 0.024; Monte Carlo permutation, mean p= 0.0014,
variance p = 0.0016 with $10^4$ permutations; Epps–Singleton test for the same distribution of two univariate samples p = 1.5 $10^{-6}$). Correcting of this analytical discrepancy has been performed on the basis that alkenone composition reflects reliably the ambient temperature during production (Conte et al., 2006). Furthermore, we allowed for a small degree of selective degradation during transport through the water column, resulting in slightly higher $SST_{UK}$ (global calibration of $U^{K'}_{37}$ from suspended
particulate matter (SPM) to ambient SST (Conte et al., 2006). By subtracting 0.094 $U^{K'}_{37}$ units, the CBeu5 $SST_{UK}$ average equals that of CBeu1–4. Application of the global calibration reveals a deviation of 0.45°C from $SST_{SAT}$ for the entire series CBeu1–5 ($SST_{SAT}$ = 0.723 $SST_{UK}$ + 5.56; φ = 35 days; $r^2$=0.60, average 21.76°C) and since the slope of the global calibration differs from our dataset, the reconstructed $SST_{UK}$ show an offset of almost +0.8°C from $SST_{SAT}$ at the lower temperatures (18°C)
and an offset of -1.9°C at 26°C. Logically, the alternative, a local calibration removes this offset. For details about this correction and the local cubic calibration we refer to supplemental material (Sup. Fig. S3).

## 2.6 Diatom data

Diatom counting has been performed as described in Romero et al., (2020). Data for *Proboscia alata* are presented here for the first time. Although *P. indica* is a weakly silicified species and subject to dissolution already in the upper waters, the proboscis (uppermost part of the frustule) is stronger silicified, easily identifiable and can be unmistakingly assigned to the corresponding species. *P. alata* is the only *Proboscia* species present in the diatom assemblage.

## 2.7 External data

### 2.7.1 Insolation

Solar insolation at 20°N has been obtained from https://www.pveducation.org/pvcdrom/properties-of-sunlight/calculation-of-solar-insolation. This record provided insolation for each 5th day. The four most important frequency components were extracted using the sum of sinusoids model of PAST4 of the insolation explaining 99.994 % of its variance have been used to generate a 5–years insolation record with daily resolution for comparison with the daily $SST_{SAT}$ record (Fig. 3).

The insolation is not a simple sinus wave, the lowest half of the insolation amplitude (7.777—10.744 W m$^{-2}$ d$^{-1}$) includes 41.2% of the year (152 d), the highest half accounts for 58.8% (213 d). The 10% highest insolation (>10.447 W m$^{-2}$ d$^{-1}$) occurs for 28% (103 d) of the year, the 10% lowest insolation values (< 08.074 W m$^{-2}$ d$^{-1}$) occur at 17% (62d) of the year (Fig. 3a).

### 2.7.2 Daily sea surface temperatures

Satellite-derived $SST_{SAT}$ values for the CBeu have been obtained through ERDDAP and are from the daily optimum interpolation (OI), AVHRR dataset (Huang et al., 2020) for 20.875N, 18.625W, the grid point closest to the location of CBeu and representing the square enclosed by 21°N, 18.75°W and 20.75°N, 18.5°W (Fig. 1, green square). Daily maps of $SST_{SAT}$ (Fig. 2) with a 0.0417° resolution have also been obtained through ERDDAP and are from the AVHRR Pathfinder 5.3 L3C dataset with 4 km resolution (Saha et al., 2018). For each cup, the daily $SST_{SAT}$ values have been averaged (binned) for the respective deployment time (each bin being the interval between the starting and ending dates for each cup deployment). Since it takes time for material produced in the water column to arrive in the

cup, the $SST_{SAT}$ have also been calculated for bins shifted by one day ($\varphi = 1$) to 140 days ($\varphi = 140$) back in time. In this study we assume that $\varphi$ is constant over time. Since the purpose of the SST proxies is to reconstruct SST, we took the $\varphi$ that gave the best correlation between the binned $SST_{SAT}$ record and the proxy-based reconstructed SST as collected by the sediment trap.

### 2.7.3 Monthly subsurface water temperatures, salinity and nutrient data

Subsurface (0–600m water depth) temperatures and salinities were obtained from the World Ocean Atlas 2018 (Locarnini et al., 2018; Zweng et al., 2019) using the statistical mean temperature on a 1° square for all decades centred at 18.5° W, 20.5° N. (Fig. 4). Monthly phosphate data for different water depths have been obtained from the World Ocean Atlas 2018 (Garcia et al., 2019) on a 5° grid point centred at 22.5° N, 17.5° W since insufficient data were available for a 1° grid point).

### 2.7.4 Wind direction and wind speed

Wind direction and strength are based on 3 hourly observations from Nouadhibou airport (20° 56′ N, 17° 2′ W; Institut Mauritanien de Recherches Océanographiques et des Pêches, Nouadhibou, Mauritania). Since these data are vector-based (direction and speed) they cannot be simply averaged (The average of 1 h 10 m/s at 350° and 1h 10 m/s at 10° is 1h 9.85 m/s at 0° and not 1 h 10 m/s 180°). Therefore, the data have been decomposed into their North–South ($V_n$) and East–West components ($V_e$), averaged and converted back to directions and speeds (see Grange 2014 and http://www.webmet.com/met_monitoring/622.html\). Since there is a high variability between the days, 11 days moving averages have been used to reduce this high frequency contribution (Fig. 3). Furthermore, the deviation from north is presented (-90° is west, + 90° east) rather than the full 360° scale).

### 2.7.5 Dust events

The dust data (Fig. 3) represent the monthly number of dust events as represented in the synoptic weather data recorded at Nouadhibou Airport (20°56′N, 17°2′W) and have been obtained from the Institut Mauritanien de Recherches Océanographiques et des Pêches, Nouadhibou, Mauritania (Romero et al., 2020) (Fig. 3f).

## 2.6 Statistics

Statistical analyses have been performed with the software package PAST4.0.4 (Hammer 2001) and with R packages 'grDevices', 'stats', 'EnvStats', 'methods' and 'car'. Phase shifts reported between proxy records and $SST_{SAT}$ represent the phase for which the correlation is highest. Phases are represented by $\varphi$, wavelengths by $\lambda$.

## 3 Results

### 3.1 Environmental data

### 3.1.1 Daily sea surface temperatures

Daily $SST_{SAT}$ are above average (21.3°C) from July to December (5 months) and low and relatively constant from December to July (7 months Fig. 3b). The maximum temperature is reached at Sept 30[th] (day 273, just over 6 months after the minimum). The daily maps of $SST_{SAT}$ for the region show that in contrast to previous years extensive upwelling continues, largely keeping away the relatively warm southern waters (such as depicted in Fig. 2d) during July to December 2007 so that the $SST_{SAT}$ are considerably reduced for this period (Fig. 3b). As a result, the annual (March 2007 to March 2008) $SST_{SAT}$ is 0.6°C lower than March 2003 to march 2007 (p= 2.1 $10^{-7}$).

The records of binned $SST_{SAT}$ averages (representing the averaged SST for the deployment period of each cup) varies depending on the assumed phase shift ($\varphi$) between the genesis of a signal at the sea surface and its arrival in the cup of the sediment trap. Minima lay between 17.9–18.4 °C, maxima between 26.2–27.2 °C and annual amplitudes between 7.8–8.9°C.

### 3.1.2 Monthly subsurface water temperatures, salinity and nutrients

The temperature–salinity diagram for the upper 600 m (Fig. 4) shows for November–December a predominantly South Atlantic Central Water (SACW) signature with relatively low salinities for a given temperature (or high temperatures for a given salinity). From January, the contribution of North Atlantic Central Waters (NACW) increases. From March to June the upper 80 m show admixture of cool, low

salinity waters (but absent in April) so that despite increasing insolation the SST stay below 19°C for this period.

For the upper 40 m a decrease in $PO_4^{3-}$ occurs from January with a minimum in April (Fig. 4). In May June and July the values are back at their February–March values and return to the April minimum from August to October, coinciding with the summer stratification. From September, values increase to the January maximum. Below 40 m, the highest $PO_4^{3-}$ concentrations are not in January but in December. Values stay more or less constant from January to May. In June and July they increase and sharply fall to a minimum in August. In September and October values are back to June values and they further increase in November and December.

### 3.1.3 Wind direction and wind speed

From 2003 to 2008 the 11 days averaged wind blows 87% of the time from NW to NE (36°E — 21°W) and 52% of the time from NNW to NNE (17°E–7°W). Most easterly directions occur in a relatively short period from November to February, when wind speed is at its lowest on average (Fig. 3d). This phase with more easterly winds was poorly developed in 2005/06. This is followed by a period of high wind speeds from March to July with winds more or less constant from the North (or NW in 2006 and 2007; Fig. 3e). A third period, from June to November, has the same overall wind as the preceding period but intercalates more westward outbreaks and wind speed tends to be lower. Over the investigated period there is a westward trend ($D = -0.0124$ d - 7.49, whereby D is the direction in degrees from W and d = Julian day since 01-01-2003; $r^2=0.14$; $p=9.45 \cdot 10^{-9}$ n = 1989). This implies that at the beginning of 2003 the average wind direction is 7.5° N and by the end of the study period (after 2000 d) this is 17.4°W. This effect is predominantly due to the years 2007 and 2008 for which from the beginning of May to the end of October the wind has a more westerly direction than during this period in the preceding years (Fig. 3). Frequency analysis shows apart from this long-term trend an annual cycle (363 d) and a 6 months cycle (184.6 d) with its most eastern direction at December 10[th].

The 11 days averaged wind speed varies between 1 and 9 m s[-1] and shows no trend (slope $1.4 \cdot 10^{-4}$). Frequency analysis shows an annual cycle (371 d) and a 6 months cycle (181 d). The phase of the cycle is 45 d (from Jan. 1[st]) with the maximum wind speed ¼ $\pi$ later, (45 + 93 = 138 d) which is May 18[th].

### 3.1.4 Dust events

The dust record shows a strong seasonal cycle ($\varphi$ = 55.33 days, $\lambda$ = 363.2 d) and the abundance of dust events lags wind speed by 10 days.

### 3.2 Sediment trap mass fluxes and upwelling species abundance

The relative abundance of upwelling species (%Upw) show a clear seasonal cycle and lag wind speed by 21 days ($r^2$=0.29, p=3.5 $10^{-9}$) and conversely dust event frequency by 11 days. The %Upw species correlates well with the $SST_{SAT}$ (negative $r^2$= 0.53 $\varphi$ = 76 p<2.5 $10^{-20}$ ; positive $r^2$=0.40, $\varphi$ = -119 p < 2 $10^{14}$). Total mass flux maxima occur in concentrated intervals of which the timing and amplitude may differ between the years. Total mass fluxes, and fluxes of carbonate, $C_{org}$, biogenic silica and the

lithogenic component all closely correlate and are synchronous (e.g. $C_{org}$ Flux = 0.065 *Total Flux +4.20 $r^2$= 0.74; Fig. 3g,h) (Romero et al., 2020). Maxima may occur at any time but are most frequent in late winter from February to March. In 2003 and 2008 a spring maximum is absent. The years 2005 and 2007 have maxima in late summer. Both the amplitude of the total mass and flux maxima differ between the years. In 2005 a series of small flux maxima occur but a large maximum is missing (Fig.

3). As a result of this irregular timing of total mass flux maxima, the most important frequency component is not the annual cycle but a cycle with a length of 257 d ($r^2$=0.17, p = 4.04 $10^{-6}$).

### 3.3 Diol fluxes

The 1,14$C_{28}$, 1,14$C_{30}$ and 1,15$C_{30}$ diols could be detected in 105 of the 120 cups. For several of these cups other diols were below the detection limit with the 1,14$C_{30:1}$ detected least, in only 61 cups. Diol

flux maxima, coincide with total mass flux maxima (Fig. 3) albeit, not every total mass flux maximum is accompanied by a diol flux maximum. The wavelength of the most important diol flux frequency component is 216 d ($r^2$=0.14, p < 3.4 $10^{-5}$). Sorting the samples in order of increasing (or decreasing) flux shows that the flux amplitude changes logarithmically with sample rank number ($r^2$=0.99). For the LDI relevant diols the flux increases (decreases) by factor two about every 10 samples (Supplementary

Fig. S4). Fluxes of individual diols show a cluster of high correlations consisting of the 1,13$C_{30}$ diol and both 1,15 diols ($r^2$ > 0.75). The 1,13$C_{28}$ diol correlates best with the 1,13$C_{30}$ diol and ($r^2$ > 0.75), less

with the 1,15 diols ($r^2 > 0.50$) and even less with the 1,14 diols. The 1,14 diols, correlate best with each other ($r^2 > 0.83$) whereby the $1,14C_{28}$ diol correlates less with the other diols than $1,14C_{30}$ diol. The $1,14C_{30:1}$ diol does not show strong correlations with the other diols, it correlates best with the $1,14C_{30}$ diol and second with the 1,15 diols. This absence of strong correlations with the $1,14C_{30:1}$ diol we relate to its low concentrations which intrinsically have a large error bar and is accompanied by occurrences below the detection limit. Excluding the $1,14C_{30:1}$ diol, the $1,13C_{28}$ and $1,14C_{28}$ diols correlate least. Thus, overall, the 14 and 15 diols correlate least ($r^2 = 0.45$), the 13 diols are intermediate but have a higher correlation with the 15 diols ($r^2 = 0.79$) than the 14 diols ($r^2 = 0.48$). We were unable to detect 1,12 diols.

### 3.4 Diol ratios and indices

The NDI, $DI_W$, $DI_R$, CDI and to a lesser extent the DSI covary to a considerable extent ($r^2 > 0.46$; Table 4, Fig. 5). The reason for this is that all have the $1,14C_{28}$ diol in the numerator, which is the most abundant diol in the record (40% of the diol flux). Furthermore, all but the NDI also have the covarying $1,14C_{30}$ (Table 3) in the numerator (second in abundance with 24% of the diol flux). Finally, in the denominator these indices all have 1,13 and/or 1,15 diols. The dominance of the $1,14C_{28}$ and $1,14C_{30}$ diols is also illustrated by the strong anticorrelation of the DCI and NDI, which have both diols in opposite positions of the fraction. Due to their annual cyclicity, statistically the diol indices correlate to environment with two maxima of opposite sign and about 181 days apart (½ cycle). For some indices the causal relation to environment is not clear a priori. In case of a negative correlation with environment the correlation is positive if shifted by ½ cycle but this is also true for the unshifted reciprocal of the proxy. Below, we often provide both and negative correlations and their phases with environmental variables without further interpretation, which will be added in the discussion.

### 3.4.1 Temperature proxies LDI, DSI and DCI

The $SST_{LDI}$ maximum of 26.7°C is 1.1°C above the binned $SST_{SAT}$ maximum which, considering the error bars is a reasonable approximation (25.6°C at $\varphi = 41$ d, Fig. 6). However, the minimum of 10.9°C is 7.7°C lower than the lowest binned $SST_{SAT}$ value (18.6°C at $\varphi = 41$). In total there are four values

below this lowest binned $SST_{SAT}$ and they occur in July 2003, July 2006, and March 2007, all in cup series CBeu1–4. A simple box plot shows that these values are outside the notches, which indicate approximately the 95% confidence interval (Fig. 7). The Rossner's Test for identifying multiple outliers (from R–package EnvStats) identifies the lowest four values (< 18°C), as outliers (p < 0.006, and for the lowest value <11°C p<0.0001). Cross correlation plots (Fig. 8) of the log transformed diol fluxes shows that the anomalies are most pronounced in the $1,13C_{30}$ diol relative to each of the 1,15 diols and do not correlate with the flux amplitude. For several samples no 1,13 diols could be detected. This lack of information automatically results in LDI values of 1, translating into 27.42°C which is unrealistic in this setting. For these reasons, such data points have been omitted.

A linear correlation of LDI to binned $SST_{SAT}$ explains 17% of the variance ($\varphi = 41$; p= $1.4\ 10^{-4}$ n=82, Table 4). The linear correlation of $SST_{LDI}$ to binned $SST_{SAT}$ also explains only 17% of the $SST_{SAT}$ variance ($\varphi = 38$; p=$1.2\ 10^{-4}$). The integrated production temperature is 23.6°C (with the outliers 22.9°C).

The DCI correlates significantly with $SST_{SAT}$ (negative correlation at $\varphi = 117$, $r^2=0.35$, p<$4\ 10^{-11}$ positive correlation at $\varphi$ =-77, $r^2=0.29$, p<$2.8\ 10^{-9}$; Fig. 6). As such $SST_{SAT}$ could be reconstructed from DCI using the positive correlation:

$$SST_{SAT} = 6.45\ DCI + 18.4\ (95\%\ conf.:\ slope\ 4.31\ to\ 8.68,\ intercept\ 17.5\ to\ 19.2) \qquad (18)$$

or the negative correlation:

$$SST_{SAT} = -8.13\ DCI + 16.79\ (95\%\ conf.:\ slope\ -9.99\ to\ -6.12,\ intercept\ 23.9\ to\ 25.8) \qquad (19)$$

The DCI lags wind strength ($\varphi = 30$ positive correlation, $r^2 = 0.29$, p<$5.5\ 10^{-9}$). The DCI shares most variance with the NDI (74%) with a negative correlation.

The DSI correlates strongly with the NDI and DCI (Table 4). The positive correlation with $SST_{SAT}$ ($r^2$= 0.25, $\varphi = 77$ p=$5.7\ 10^{-5}$) is much lower than the negative correlation ($r^2$= 0.42, $\varphi = -124$ p=$3.4\ 10^{-8}$). For a considerable number of samples no unsaturated diols could be detected in which case the DSI is not

defined and 1. However, if the response of the combined 1,14 diols is 100 times higher than the detection limit of the $1,14C_{30:1}$ diol, the DSI has to be $>0.99$ otherwise it would have been detected. If the response is less than 100 times, it may be that the DSI was $< 0.99$ but the diol concentrations were in general too low to detect the $1,14C_{30:1}$. Taking also into account the different response factors of the diols, 11 samples without detected unsaturated diols were identified for which the DSI could be $>0.99$. Statistical analyses including these samples showed mostly slightly lower correlations compared to the dataset without them and we present the results with these 11 samples excluded from the DSI.

### 3.4.2 Upwelling and upper ocean productivity proxies DIR, DIW and CDI

The $DI_R$, as a proxy for the *Proboscia* contribution shows correlates significantly with the ln *P. alata* flux ($r^2 = 0.29$, $p = 1.6\ 10^{-4}$) but not with the total mass flux or its logarithm (Fig. 9).

The $DI_W$ as a proxy for upwelling correlates significantly with the ln *P. alata* flux ($r^2 = 0.12$; $p = 0.024$) but not with the total mass flux ($r^2 = 0.011$, $p = 0.25$). The correlation of the $DI_W$ with the LDI is negative low ($r^2=0.04$) and just not significant ($p=0.09$), for the $DI_R$ and LDI the correlation is also negative but much higher ($r^2=0.24$ $p=6.3\ 10^{-7}$). Both the $DI_R$ and $DI_W$ show a negative correlation with wind strength which is best at $\varphi = 79$ (Table 4).

The Combined Diol Index (CDI) (Rampen et al., 2014a), is almost identical to the $DI_R$ ($r^2=0.995$) and therefore has not been evaluated separately.

### 3.4.3 Upper ocean nitrate and phosphate concentration proxy NDI

The NDI, shares about half its variance with the CDI, $DI_W$ and $DI_R$, even higher with the DCI (74%) and DSI (67%). The NDI shows neither correlation with *P. alata* flux ($r^2=0.005$, $p=0.6$) nor with the total mass flux ($r^2=0.0014$, $p=0.7$), but correlates well ($r^2= 0.38$ $p=4.1\ 10^{-12}$) with the relative abundance of upwelling indicating species (%Upw) in the samples.

### 3.5 Alkenone based $U^{K'}_{37}$ and $SST_{UK}$

The $SST_{UK}$ values are 18.0–26.1°C and best fit to $SST_{SAT}$ if $SST_{UK}$ at $\varphi = 35$ days (Fig. 6). For this phase lag the binned $SST_{SAT}$ are 18.5–26.1°C and the $SST_{UK}$ record thus has about the same amplitude extending 0.5°C below the minimum and equalling the maximum $SST_{SAT}$. The measured $U^{K'}_{37}$ explains

59% of the $SST_{SAT}$ variance and the $SST_{UK}$ 60%. The integrated production temperature (flux corrected) is 21.14°C, nearly the $SST_{SAT}$ for φ = 35 (21.37°C).

The regressions of $U^{K'}_{37}$ and $SST_{UK}$ to binned $SST_{SAT}$ do not differ much in the explained variance and
phase (φ 35 d, $r^2$ =0.59 *vs*. 0.60). The flux corrected $SST_{UK}$ for CBeu1–5 prior to correction is 21.54, after correction it 21.14°C which is only 0.23°C lower than the $SST_{SAT}$ for the same period (21.37 φ=35) and well within the standard error of the method (1.2°C) (Conte et al., 2006).

## 4 Discussion

### 4.1 Water temperatures during the sampling period

During the entire record, $SST_{SAT}$ tend to remain relatively constant for periods of one to three weeks with rapid shifts of 1–2°C between them (Fig. 3). During summer these shifts tend to be larger and they may be up to 4.5 °C within a week at the transitions between summer and winter (in 2004). The duration of these 1–3 weeks periods of stable temperatures is similar to the deployment of individual sediment trap cups (8–24 days). As a result, these periods are partly reflected by, and consistent features
of, the binned $SST_{SAT}$ record. This is much more apparent after November 2006 when the cup deployment is systematically less than 10 days and close to the duration of the shorter-term events of the system. This demonstrates that to capture the systems dynamics, these shorter deployment times are the preferred mode of operation, if not a requirement.

The $SST_{SAT}$ does not simply follow the insolation curve, but a substantial rise in temperature only starts
from the beginning of June, only a few weeks before the insolation maximum. We attribute the delay to maximum intensity of the trade winds during late winter and spring, intensifying upwelling and pumping cool, low salinity and nutrient-rich SACW waters to the surface along the coast and subsequently spread these westward over the trap site. Wind speeds are strongly reduced and upwelling is at a minimum for July to October, enabling the surface waters to strongly increase in temperature,
except for 2007 when upwelling reduces much less and summer temperatures stay relatively low. The temperature gradient is always small for the upper 20 m, the permanently mixed layer.

The most prominent feature of the monthly water temperature profiles is the strong annual cycle at the sea surface, which becomes smaller with increasing depth. In the upper 100 m the relatively short distances between the temperature profiles for different depths (the strength of the temperature gradient) are clearly smaller from January to June compared to the rest of the year. We attribute this to mixing and upwelling. This seems to be most intense in May and June. In May, elevated temperatures up to 400 m depth indicate this and in June the upwelling and mixing result in SST that are even lower than the preceding months, despite stronger solar insolation. We also observe that the highest temperatures at depth (80–200m) lead the SST by about three months (Fig. 4) which in case of temperature proxies generated at these depths could lead to proxy-derived temperature records leading SST.

## 4.2 The annual cycle, wind, dust, SST and upwelling

All environmental parameters investigated show a dominant annual cycle modulated by a semi-annual component. This is also true for most of the proxy records. This implies that a significant correlation of a proxy parameter to an environmental variable, with a given phase shift has a high chance to also provide significant correlations with other environmental parameters, albeit with different phase relations. Although is no problem for proxy records with a known causal relation with environment, such as the $U^{K'}_{37}$ with temperature, this becomes a problem for proxies where this relation is less clear. First, it is not clear which environmental parameter is the forcing factor since if one parameter shows a significant correlation, the others will do as well. Second, the phase relation is unclear since two correlation maxima occur each environmental cycle albeit with opposite sign. Additional arguments, e.g., from the data structure and/or functioning of the system are needed to decide which proxy–environment relation is most likely to be causal. In our system, off Cap Blanc, the modulation of the semi-annual frequency component causes asymmetry in the cycles of the environmental variables (Fig. 3). As a result, best correlation and anticorrelation are unequal and their phases not ½ cycle apart. Comparison of wind speeds and $SST_{SAT}$ in this region shows that both have a short period of high values, followed by a relatively long period of low values. If proxy records follow environmental change closely, they also should display this asymmetry in the annual cycle. High wind speeds in late winter and spring drive upwelling, preventing $SST_{SAT}$ to increase with solar insolation so that only

when wind ceases by the beginning of June, $SST_{SAT}$ can rapidly rise. This results in maximum wind speed leading maximum SST by 122 days (based on 11 day means, $r^2=0.38$) and minimum SST leading maximum wind speed by 91 days ($r^2=0.37$). The asymmetry in the annual cycle thus leads to an alternation of 122+91=213 day difference between correlation optima, followed by a 365-213=152 day difference between the next correlation optima. We expect to see this asymmetry also in the other proxy records. The frequency of dust events lags wind speed by 10 days so that most events occur during the period of most active upwelling. Due to this small phase difference in relation to the generally two to three times longer sample frequency of the sediment traps it is impossible to separate the effects of upwelling and dust input on the export flux, species and lipid composition. Since wind speed is driving both upwelling and dust event frequency, only relations to wind speed are further considered.

### 3.3 Diol fluxes

Diol flux maxima, like total flux maxima, may occur in any season and, therefore, the record is not dominated by an annual cycle (Fig. 10). Whereas all diol maxima occur during total flux maxima the opposite is not true. Apparently, flux maxima may follow from different environmental configurations, whereby some are accompanied by high diol export. An obvious hypothesis in this region would be that both productivity and dust events may induce high export production and that dust events not necessarily occur when diol concentrations are high. Unfortunately, we do not have sufficient data to support this hypothesis. Correlations between fluxes of individual diols show that the 1,14 and 1,15 diols correlate least whereas the 1,13 diols are intermediate but correlate better to the 1,15 diols. Indeed, flux maxima of 1,14 diols are partly independent of those of 1,15 and 1,13 diols. This supports earlier work suggesting different sources for 1,15 and 1,14 diols (e.g. Rampen et al., 2007; Gal et al., 2021), whereby the 1,13 diols may be derived from both sources. The logarithmic relation between the diol flux reached and its frequency (many low fluxes, few very high) is important for the diol composition and diol proxy values integrated over timescales longer than a cup–to–cup basis. The more uneven the distribution, the more influence a few large fluxes have on the total value. In our case, the average difference between a given flux and next largest flux is about 6.5% so that on average the 10[th] largest flux still has half the influence of the largest one (see also 4.5).

## 4.4 Diol based temperature proxies LDI, DSI and DCI

The LDI uses the underlying assumption that the percentages of 1,13 diols (relative to the $1,15C_{30}$ diol) decrease with temperature. For the $SST_{LDI}$ outliers (values below 18°C) we could not find any analytical explanation. These outliers arise from excess production of both 1,13 diols. The occurrence of these anomalies suggests that this excess production is unrelated to temperature. We observe that all three events with excess 1,13 diols (and very low LDI values) occur during Total Flux maxima (compare Figs. 5, 9) and as such we speculatively suggest that factors leading to elevated export productivity play a role. However, also with the outliers omitted, we observe a poor relationship between LDI and $SST_{SAT}$ and a phase lag of 41 days ($r^2$=0.17) but still a significant seasonal cycle ($\lambda$=347 d, $r^2$=0.36, p=1.2 $10^{-10}$). This contrasts with trap studies from the equatorial Atlantic, Mozambique Channel and to a lesser extent the Cariaco Basin where a general absence of a seasonal cycle in the LDI has been observed (de Bar et al., 2019). The $SST_{LDI}$ integrated production temperature (23.6°C) is well above the annual $SST_{SAT}$ (21.3°C). This may be explained by the general observation that 1,13 and 1,15 diol production is weighted towards the non-upwelling season, which has above average temperatures (as is also indicated by the poor correlation of 1,15 and 1,13 diols with 1,14 diols; Fig. 8).

Correlation is not significant between LDI and DCI (p=0.89), and only just between LDI and DSI (p=0.045). The LDI is based on 1,15 and 1,13 diols. The DSI and DCI on 1.14 diols and as such they are mathematically independent. If the DSI or DCI also reflect temperature, their 1,14 diols must have been produced in a temperature regime independent of that recording the 1,15 diols.

Due to the partial absence of detected unsaturated diols, valid values of the saturation index DSI could be calculated for only half of the samples. Application of the culture-based transfer function to these samples provides unrealistic water temperatures for the CBeu surface waters (range 10.1–41.0°C, average 33.2°C) and this transfer function is thus not applicable to this region. The DSI lags $SST_{SAT}$ by 76 d ($r^2$= 0.25, n=58), which is a much more than observed for the LDI ($\varphi$ = 41). This difference may indicate that sinking speed of the 1,14 diols (comprising the DSI) differs from that of the 1,13 and 1,15 diols (comprising the LDI). It may another argument that changes in the 1,14 diol composition represent an environmental response different from the response changing the 1,13 and 1,15 diols composition. One explanation may be that the 1,14 diols are produced by different organisms than the 1,13 and 1,15

diols, which allows for both different sinking speeds and different environmental responses. In the marine environment, the 1,13 and 1,15 diols are predominantly derived from marine eustigmatophytes (e.g. Gelin et al., 1997; Volkman et al., 1992, 1999; Versteegh et al., 2000; Rampen et al., 2014b). The $1,14C_{32}$ diol is a major diol in the diatom *Apedinella radians* (Rampen et al., 2011) but since we do not observe this diol in our cup samples, we infer that this species does not significantly contribute diols in our case. In our samples we do observe a relative good correlation between fluxes of the diatom *Proboscia alata* with the 1,14 diols, despite possible other factors such as diatom dissolution or contribution of 1,14 diols by, as yet unknown, other sources (31% explained variance, Fig. 9). This correlation is considerably higher than the correlation between *P. alata* fluxes and 1,13+1,15 diols (4%) or between total diatoms and 1,14 diols (17%). This suggests that *P. alata* as the only *Proboscia* species encountered, significantly contributes 1,14 diols whereas its 1,13+1,15 diol contribution is insignificant. Culture experiments demonstrate this ability for *P. alata* to produce 1,14 diols (Sinninghe Damsté et al., 2003). Our observations also agree well with diol data from sediment traps from the Arabian Sea where abundance of 1,14 diols covaries with upwelling and appears unrelated to the abundance of 1,15 diols (Rampen et al., 2007). Also a recent sediment trap study from the East Sea (Gal et al., 2021) supports the hypothesis of 1,14 diols being produced by a different plankton population than the 1,13 and 1,15 diols. In these sediment traps (March 2011–March 2012) the 1,14 diols show very similar and temporally narrowly confined flux maxima in Oct. and Nov. 2011 whereas the 1,13 and 1,15 diols show a very different pattern with broader flux peaks and almost permanent presence.

Assuming the upwelling related *P. alata* being a major 1,14 diol source in our research area, it makes sense to investigate if and how the DSI relates to wind induced upwelling and %Upw since stronger upwelling induces lower temperatures in the photic zone. During the investigated time interval, $SST_{SAT}$ lags wind speed by about one season ($\varphi$ =122 $r^2$=0.342). With diatom sinking rates in this region of 100 – 250 m d$^{-1}$ (Fischer and Karakaş 2009) and 1280 m trap depth, diatoms are expected to arrive at the trap within 5–12 days after starting to sink. We observe that the abundance of upwelling species (%Upw) lags wind speed by 21 days ($r^2$=0.26, p=6.8 10$^{-9}$) agreeing reasonably well with the estimated sinking speed (Romero et al., 2020). We therefore expect the DSI to behave similar to the %Upw lagging the wind speed by 21 days and leading $SST_{SAT}$ by 101 days. We observe the DSI lags the

SST$_{SAT}$ by 76 days and is not leading it as was assumed based on the arguments above. However, the inverse DSI (higher values more diol unsaturation) provide a completely different picture lagging wind speed with the same phase ($\varphi = 21$, $r^2=0.24$ p=9.7 10$^{-5}$) and correlates well with %Upw ($r^2=0.39$). It also

correlates surprisingly well with SST$_{SAT}$ leading it by 124 days ($\varphi=-124$, $r^2=0.$ 42). We therefore, propose that the inverse DSI reflects upwelling related changes with a higher contribution of unsaturated diols upon stronger upwelling. Since upwelled waters are cool, upwelling intensity is inversely correlated with SST. Thus, the degree of 1,14 diol unsaturation reflected by the DSI could also reflect upwelling temperature. As discussed above, in this region CBeu, upwelling has a strong

influence on the annual SST curve with solar insolation determining SST only in summer when upwelling is weak. Furthermore, 1,14 diols seem to be primarily produced by *Proboscia*, tied to upwelling, or higher nutrient levels. As such, the DSI may primarily reflect upwelling induced temperature effects and only secondary the temperatures resulting from direct solar radiation. Interestingly, also in the East Sea sediment trap (Gal et al., 2021) the 1,14C$_{28:1}$ is the only diol involved

in the diol flux maximum in (cool) spring 2011 whereas all 1,14 diols are involved in the narrow diol flux peak in (warm) autumn 2011. Both these spring and autumn diol maxima occur during productivity maxima which also show maxima of diatom export production (Kim et al., 2017). Gal et al. (2021) ascribe the spring 1,14C$_{28:1}$ diol maximum to *P. alata* whereas they relate the autumn maximum including also the 1,14C$_{30}$ diols to *P. indica,* a species absent from Cap Blanc. The DSI thus seems to be

heavily modulated by processes related to high productivity. Since high productivity in the ocean is temporally and spatially highly variable, a possible relation to longer-term SST$_{SAT}$ averages may be less straightforward and a high-resolution (sub)weekly monitoring of environment and diol production may be needed better understand the environmental information carried by the 1,14 diol record.

The DCI leads the SST$_{SAT}$ ($\varphi=-77$, $r^2=0.29$). Since the reconstructed SST cannot precede the actual

SST, it is unlikely that the DCI reflects SST. Moreover, available transfer functions (see Methods 3.4) lead to unrealistic temperatures for CBeu. The inverse of the DCI lags SST (best correlation $r^2=0.35$ at $\varphi=117$) but simultaneously its average chain length decreases with temperature, which is opposite to the common physiological response of organisms to temperature (higher lipid melting point/viscosity at

higher temperatures). Consequently, a causal relation through physiological adaptation of the diol-producing organisms to SST must be considered unlikely.

The DCI follows wind speed by 27-31 days ($r^2$=0.28, $\varphi$=27, p= 7.5 $10^{-9}$; $\varphi$=31, p=6.4 $10^{-9}$) and upwelling species (%Upw) by 6–9 days ($r^2$=0.42). This suggests that like the DSI, also the DCI reflects changes in the upwelling regime. Correlations are not significant between DSI or DCI and *P. alata* fluxes. Interestingly, both these supposedly upwelling related DSI and DCI show higher correlations to $SST_{SAT}$ than the temperature proxy LDI. We attribute this to the (annual) cyclic nature of the system and strong influence of upwelling.

### 4.5 The integrated production temperature: $SST_{LDI}$ over longer time intervals.

The sediment trap intercepts the material that would continue on its way to the ocean floor 1500 m below the trap. Although the signal can be distorted on its way further to the ocean floor by (selective) degradation and transport, sediment traps still provide important insight in the evaluation of the sedimentary signal. Time series from the sediment record mostly have a resolution of years to millennia, much lower than the resolution of the CBeu trap series. To get a better insight in the effect of the export dynamics as observed for CBeu on proxy values representing longer time periods, the values obtained from the individual trap cups have been weighed by the respective fluxes. For the LDI, the integrated production temperatures ($IPT_{LDI}$) have been calculated for a 19–cup moving window on the basis of the LDI diol fluxes and the $SST_{LDI}$, both with and without the $SST_{LDI}$ outliers (Fig. 11). The LDI diol fluxes have also been combined with the $SST_{SAT}$ at $\varphi$=41d (the phase relation with the best correlation). The thus obtained $IPT_{SAT}$ represents the ideal case the LDI perfectly records the $SST_{SAT}$. The $IPT_{SAT}$ values from January 2004–September 2005 (IPT 4 to 23) are higher than the moving average $SST_{SAT}$ values due a single flux maximum in early December 2004. Considering a $\varphi$=41d, this production peak records high $SST_{SAT}$ in early autumn 2004. Thus, even if the trap cups would perfectly reflect $SST_{SAT}$, a single flux peak may cause the IPT to differ considerably from the $SST_{SAT}$ mean over the same period. From Autumn 2005–Spring 2008 the $IPT_{SAT}$ values reflect $SST_{SAT}$ as a result of a more constant flux magnitude.

It is obvious that even if the LDI is corrected for diol flux data ($IPT_{LDI}$ with, and without outliers), values are considerably higher than what should be reflected in the sediments ($IPT_{SAT}$) and they are similar to summer temperatures. In the CBeu record the high temporal resolution enables identification of outliers. With a much lower resolution (e.g. a sediment sample taken below the trap) these would be integrated in the signal. The differences between the $IPT_{LDI}$ with, and without outliers demonstrate the

effect of the outliers on the $IPT_{LDI}$. Especially the outliers at cups centred at 204d (10.9°C) and 1530d (14.9°C) have a considerable impact since they combine low reconstructed temperatures with significant fluxes (Fig. 11). We thus demonstrate that single high fluxes and outliers in the data can have large effects on the sediment signal and this may be one reason why higher productivity regions such as upwelling areas there correlation is less good between $SST_{LDI}$ values in sediment surface

samples and annual mean $SST_{SAT}$.

## 4.6 Diol based productivity/nutrient concentration proxies NDI, DIR, DIW, and CDI

### 4.6.1 NDI

The NDI, DSI and DCI correlate well ($r^2 > 0.68$). This is expected since although the NDI consists of all diols encountered, it combines the DSI and inverse DCI. The 1,14 diols in these two latter proxies

obviously dominate the NDI. The NDI has been proposed to reflect $NO_3^-$ and $PO_4^{3-}$ concentrations in the surface waters (Gal et al., 2018, 2019, 2021). If in our setting, we assume that upwelling and the relative abundance of upwelling species (%Upw) are associated with nutrient-rich conditions at the CBeu site, a significant positive correlation between NDI and %Upw is expected. However, our results show a negative correlation (Table 4). The interpretation of the NDI as reflecting $NO_3^-$ and $PO_4^{3-}$

concentrations in the surface waters is further complicated by a clear annual cycle in our dataset whereas monthly $PO_4^{3-}$ and $NO_3^-$ concentrations show no cyclicity. Furthermore, mostly concentrations of these nutrients show no correlation with the NDI (p <0.04).

A closer look at the NDI reveals that, the slopes of the respective transfer functions relate to each other according to the Redfield Ratio N:P = 16:1. This may be expected since the original calibration between

NDI and nutrients (Gal et al., 2018) is primarily based on sediment-derived NDI values and photic zone

annual nutrient levels (n=216), the latter following the Redfield ratio. SPM summer samples with associated summer nutrient concentrations are also included in this Gal et al. (2018) sediment–sea surface dataset. The NDI–$PO_4^{3-}$ relation of these SPM samples falls within that of the sediment samples but, for the SPM samples alone, correlation with the $PO_4^{3-}$ concentrations is not significant ($r^2=0.2$,

p=0.14). A sediment trap study from the East Sea (Gal et al., 2021), shows a different relation between the trap NDI values and monthly sea surface $PO_4^{3-}$ concentrations ($r^2=0.59$, n=26). In the East Sea, $PO_4^{3-}$ concentrations and NDI show a clear annual cycle. However, this is true for most environmental parameters, and it would be interesting to see what part of the observed correlation between NDI and $PO_4^{3-}$ is causal and what part results from a common (annually cyclic) forcing factor.

**4.6.2 $DI_W$, $DI_R$ and CDI**

The $DI_w$, $DI_R$ and CDI have the same 1,14 diols in the numerator and only differ in the combination of 1,13 and 1,15 diols in the denominator. The CDI only adds the $1,15C_{30}$ diol to the $DI_R$. The $DI_w$ has been directed as a proxy for the contribution of *Proboscia* diols relative to other diols. In our dataset, its correlation with the *Proboscia* fluxes is low but still significant ($r^2=0.12$ p=0.024). The proxies for

upwelling strength $DI_R$ and CDI show even higher correlations with *Proboscia* (Table 4).

The $DI_R$ has been suggested to be proxy reflecting total export productivity. We observe high values (>0.9) of this proxy. This would agree with the observation that this region experiences permanent upwelling. Both the $DI_W$ and $DI_R$ correlate negatively with wind speed (at $\varphi = 79$) and $SST_{SAT}$ (at $\varphi = -69$). Closer observation shows that generally, all diol fluxes more or less covary except in 2007 where

pulses in 1,13 and 1,15 diols precede those of the 1,14 diols (Fig. 8) and the $DI_R$ sinks below 0.8. Comparison to the total mass flux (correlation insignificant) and to the %Upw reveals that this change in diol composition is not accompanied by a consistent change in export flux so that an explanation for the relative increase in 1,15 diols seems to require a more subtle knowledge of the relation between diol abundance and environment on the CBeu trap data alone.

Other trap studies reporting diol fluxes from tropical Atlantic are upper traps from 1150 m depth in the Eastern Atlantic near the Guinea Dome and influenced by seasonal upwelling (M1U), from 1235 m depth in the oligotrophic Central Atlantic (M2U) and from 1130 m depth in the western Atlantic and

under seasonal influence from the Amazon outflow (M4U) (de Bar 2019). It appears that for the 1,13+1,15 diols our average of 3.9 µg m$^{-2}$d$^{-1}$ for CBeu5 is only 1.5 times the average for the nearest trap M1U (2.6 µg m$^{-2}$d$^{-1}$), and about 3 times the flux of the oligotrophic central Atlantic M2U (1.2 µg m$^{-2}$d$^{-1}$) but only half that of the westernmost M4U (7 µg m$^{-2}$d$^{-1}$). For the 1,14 diols the average of 1.7 µg m$^{-2}$d$^{-1}$ for CBeu5 is least three times higher than for the other traps (0.5 for M1U, 0.01 for M2U and 0.3 for M4U). Since CBeu5 is the only site under permanent upwelling influence, we infer that the 1,14 diols are particularly abundant in the (coastal) upwelling whereas the 1,13 and 1,15 diols also increase with increasing productivity but seem to be less bound to upwelling and possibly better reflect the more oligotrophic open ocean (agreeing with Versteegh et al., 2000; Rampen et al., 2008). Seen in this light, it may be expected that at CBeu the SST$_{LDI}$ reflect summer SST$_{SAT}$ since the oligotrophic open ocean conditions are largely constrained to the short (July–October) period with high temperatures.

## 4.7 U$^{K'}_{37}$ and SST$_{UK}$

The 35 days phase lag of SST$_{UK}$ relative to SST$_{SAT}$ suggests that it takes about one month between fixation of the alkenone composition in the cell and the collection of the alkenones in the cup of the sediment trap which implies an average sinking rate of 38 m d$^{-1}$.

The complex cubic transformation of U$^{K'}_{37}$ values to SST$_{UK}$ for SPM derived from a global dataset (Conte 2006) has a lower correlation than U$^{K'}_{37}$ itself. This implies that in this case a linear transformation would have performed equally well. This we may expect since our data cover only a fraction (18–27°C) of the global temperature range and for this fraction the cubic transformation behaves almost linear. Since the explained variance is much lower (60%) than for the global calibration (97%) we infer that other factors than surface water temperature influence the alkenone composition. If the reconstructed SST$_{UK}$ would perfectly project to the SST$_{SAT}$ the slope of the regression would be 1 and the intercept 0. However, the slope of the SST$_{UK}$ regression to binned SST$_{SAT}$ is 0.72 and the intercept 5.6, which implies that the global SPM calibration overestimates local SST$_{SAT}$ below 20°C and underestimates local values above this temperature. If we assume that the global calibration is correct and the alkenones reflect the water temperature the organisms were subject to, the conclusion would be that above 20°C the organisms live in water that is slightly cooler than that at the sea surface (SST$_{SAT}$)

and slightly warmer below this temperature. This could be explained by a higher influence of solar irradiation at the sea surface than at the subsurface. This is only feasible if the alkenone-producing organisms live partly or entirely below the surface mixed layer. Surface currents are from the NE where (upwelling) SST is lower. Therefore, admixture of alkenones produced in upstream could also be an explanation. Alternatively, we may adjust the SPM calibration of Conte et al. (2006) such that we obtain

a local calibration where the slope and intercept agree with the binned $SST_{SAT}$ data ($\varphi = 35$) providing (see also supplementary Fig. S3):

$$SST_{UK} (\degree C) = 0.64733 + 54.293 \, U^{K'}_{37} - 52.894(U^{K'}_{37})^2 + 23.382 \, (U^{K'}_{37})^3 \qquad (20)$$

with $\varphi = 33$ and $r^2 = 0.54$. Just like the global calibration of Conte et al. (2006) this regional calibration shows a higher correlation to $SST_{SAT}$ than any of the diol proxies discussed above.

## 5 Conclusions

The variation in diol, fluxes and relative abundances as observed in sediment trap cups off Mauritania from 2003 to 2008 have been compared with environmental conditions, alkenone and plankton

composition for the same region and time period. From this comparison, it appears that:

1. Peak total mass fluxes of material to the sediment trap do not show a statistically significant annual cycle but may occur throughout the year. Nevertheless, total mass flux maxima are most abundant during spring. We explain this rather unpredictable occurrence of these flux maxima by attributing them to result from the passage of upwelling filaments over the sediment trap, which occurs most often

during spring, but is not limited to this.

2. Off Cap Blanc, upwelling variability is the major environmental variable. It shows a strong annual cycle in response to the strength of the trade winds. Sea surface temperature also shows a strong annual cycle remaining low in winter and during vernal upwelling and following insolation when upwelling is reduced during summer. It lags upwelling by 130 days. As a result of the predominant annual cycle in

both temperature and upwelling, correlations between parameters and/or proxy records should be

interpreted with care and phase relations should be considered to identify the most likely forcing mechanism.

3. The alkenone-based $U^{K'}_{37}$ lags SST by 35 days and excellently reconstructs both SST with respect to amplitude and absolute values.

4. The diol derived LDI lags SST by 41 days. It correlates only weakly with SST. On average the $SST_{LDI}$ reflects the high $SST_{SAT}$ prevalent during the more oligotrophic summer period. The LDI shows several outliers to very low reconstructed temperatures, which we attribute to an additional source of 1,13 diols and which have a considerable influence on the low resolution $SST_{LDI}$ signal as may be encountered in the underlying sediment.

5. The diol-derived nutrient index NDI, the DCI as well as the percentage of upwelling species show a higher correlation to SST than the LDI. However, they lead the SST by several months and their variability is most likely a response to upwelling associated processes such as a reduction in temperature, increased nutrient content and modified species composition. A rather intriguing result is the anticorrelation between the diol-derived nutrient proxy NDI and upwelling intensity.

## 6 Code availability

Not applicable

## 7 Data availability

Data are available at https://doi.org/10.1594/PANGAEA.940305 (Versteegh et al., 202X).

## 8 Author contribution

GJMV interpreted the data and wrote most of the manuscript. KAFZ discussed and edited the paper prior to submission. JH processed performed the lipid analyses of CBeu5 and contributed to the Material and Methods section. OER contributed the diatom data and together with GM critically reviewed earlier versions of the manuscript. GF coordinated the sediment trap project.

## 8 Competing interests

The authors declare that they have no conflict of interest.

## 9 Disclaimer

Publisher's note: Copernicus Publications remains neutral with regard to jurisdictional claims in published maps and institutional affiliations.

## 10 Acknowledgements

We thank Eleonora Uliana for measuring the diol data of CBeu1–4 and Enno Schefuss for constructive comments on an earlier version of the manuscript. We thank N.N. and N.N. for their constructive reviews.

## 11 Financial support

This research has been financially supported by the German Science Foundation (DFG) through grant
GZ:EXC 2077/1 Further financial support was obtained from the Helmholtz Association (Alfred Wegener Institute Helmholtz Centre for Polar and Marine Research). The article processing charges for this open-access publication were covered by Bremen University

## 12 Review statement

This paper was edited by N.N. and reviewed by N.N. and N.N.

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

Table 1. Deployment data of sediment trap CBeu

| Mooring | Coordinates | GeoB No. cruise | Trap depth (m) | Ocean Bottom depth (m) | Sample amount | Capture duration (sample no. = days) | Sampling interval |
|---|---|---|---|---|---|---|---|
| 1 | 20°45'N 18°42'W | - POS 310 | 1296 | 2714 | 20 | 1=10, 2–20=15.5 | 5 Jun 2003– 5 Apr 2004 |
| 2 | 20°45'N 18°42'W | 9630-2 M 65-2 | 1296 | 2714 | 20 | 1–20=22 2–19=23 | 18 Apr 2004 – 20 Jul 2005 |
| 3 | 20°45.5'N 18°41.9'W | 11404-3 POS 344-1 | 1277 | 2693 | 20 | 21.5 | 25 Jul 2005 – 28 Sept 2006 |
| 4 | 20°45.7'N 18°42.4'W | 11835-2 MSM 04b | 1256 | 2705 | 20 | 1=3.5 2–20=7.5 | 28 Oct 2006 – 23 Mar 2007 |
| 5 | 20°44.9'N 18°42.7'W | 12910-2 POS 365-2 | 1263 | 2709 | 38 | 1,2=6.5 3–38=9.5 | 28 Mar 2007 – 17 Mar 2008 |

Table 2. Composition and position of the diols in the indices.

| Index \ Diol | $1,13C_{28}$ | $1,13C_{30}$ | $1,15C_{30}$ | $1,14C_{28}$ | $1,14C_{30}$ | $1,14C_{28:1}$ [1] | $1,14C_{30:1}$ |
|---|---|---|---|---|---|---|---|
| LDI | D | D | N | | | | |
| DSI | | | | N | N | D | D |
| DCI | | | | D | N | | |
| $DI_R$ | | | D | N | N | | |
| $DI_W$ | D | D | | N | N | | |
| CDI | D | D | D | N | N | | |
| NDI | D | D | D | N | D | N | D |

Since all proxies are indices, all diols in the numerator (N) are also in the denominator (D).
[1]The $1,14C_{28:1}$ diol has not been detected in the CBeu1−5 samples


Table 3: Correlations of diol concentrations and their natural logarithms for CBeu1–5.

|  | | linear correlations[1] ($r^2$) | | | | | | |
|---|---|---|---|---|---|---|---|---|
|  | Lipid | $1,13C_{28}$ | $1,14C_{28}$ | $1,13C_{30}$ | $1,14C_{30:1}$ | $1,14C_{30}$ | $1,15C_{30}$ | $1,15C_{32}$ |
|  | $1,13C_{28}$ | — | *0.35* | *0.59* | *0.01* | *0.31* | *0.44* | *0.58* |
|  | $1,14C_{28}$ | 0.42 | — | *0.14* | *0.01* | *0.69* | *0.11* | *0.20* |
|  | $1,13C_{30}$ | **0.75** | 0.29 | — | *0.01* | *0.19* | *0.70* | ***0.81*** |
| natural | $1,14C_{30:1}$ | 0.18 | 0.17 | 0.14 | — | *0.20* | *0.00* | *0.01* |
| logarithm | $1,14C_{30}$ | 0.58 | **0.83** | 0.50 | 0.33 | — | *0.12* | *0.20* |
| correlati- | $1,15C_{30}$ | 0.62 | 0.36 | **0.82** | 0.14 | 0.51 | — | ***0.84*** |
| ons[1] ($r^2$) | $1,15C_{32}$ | 0.70 | 0.34 | **0.84** | 0.17 | 0.51 | **0.85** | — |
|  | 1,13 vs 1,14 | 0.48 | | | | | | |
|  | 1,13 vs 1,15 | **0.79** | | | | | | |
|  | 1,14 vs 1,15 | 0.45 | | | | | | |

Outliers: $1,13C_{28}$ and $1,13C_{30}$ of CBeu1-4, CBeu3-18, CBeu4-18 and CBeu4-19 excluded. Bold typeface, correlations >0.74.

[1]Linear correlations in italics.


Table 4: Correlations ($r^2$) between diol proxies and environment

| | $DI_W$ | $DI_R$ | CDI | NDI | DCI | DSI | LDI |
|---|---|---|---|---|---|---|---|
| $DI_R$ | **0.65** | | | | | | |
| CDI | **0.69** | **1.00** | | | | | |
| NDI | **0.49** | **0.46** | **0.48** | | | | |
| DCI | *0.32* | *0.15* | *0.16* | *0.74* | | | |
| DSI | 0.080 | *0.0004* | 0.0005 | **0.67** | *0.46* | | |
| LDI | *0.036* | *0.27* | *0.24* | *0.029* | *0.0003* | **0.09** | |
| ln *P. alata* Flux | **0.12** | **0.29** | **0.29** | 0.000 | 0.051 | *0.019* | |
| Total Flux | 0.011 | 0.024 | 0.021 | 0.001 | *0.017* | *0.013* | |
| % Upw spp. | *0.20* | *0.07* | **0.09** | *0.38* | 0.42 | *0.39* | *0.02* |
| Wind Speed | *0.19 φ 79* | *0.11 φ 79* | **0.093 φ 35** | *0.28 φ 32* | **0.29 φ 30** | *0.24 φ 21* | |
| SST (lead) | *0.23 φ -69* | *0.10 φ -69* | *0.13 φ -69* | *0.38 φ -77* | **0.29 φ -77** | *0.42 φ -124* | **(0.21 φ 140?)** |
| SST (lag) | **0.27 φ 94** | **0.13 φ 105** | **0.15 φ 104** | **0.34 φ 102** | *0.35 φ 117* | **0.25 φ 76** | **0.17 φ 41** |

Correlations with p<0.05 in bold, negative relations in italics. LDI outliers not included



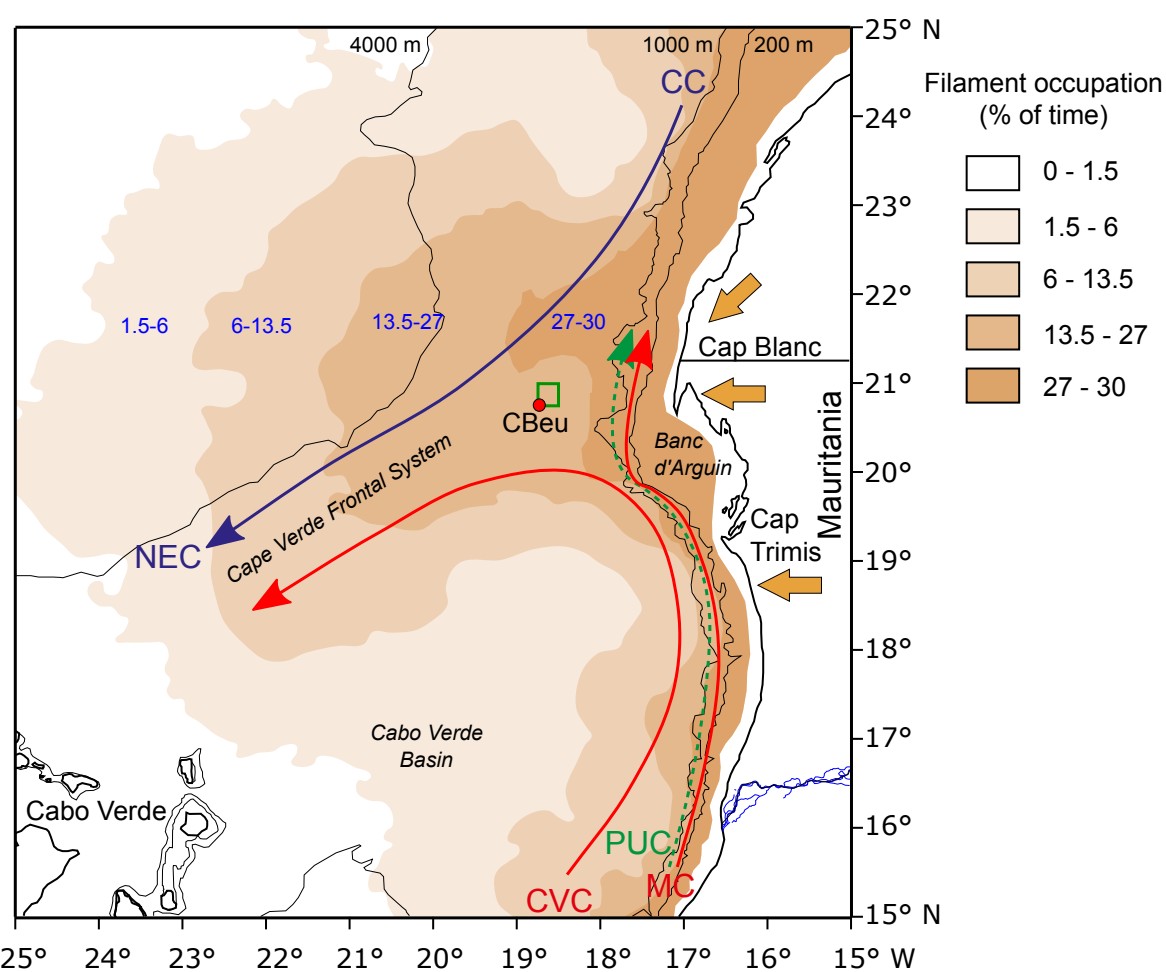

Figure 1: The Mauritanian upwelling system. CC, Canary Current; NEC, North Equatorial Current; CVC, Cap Verde Current; MC, Mauritanian Current; PUC, Poleward Under Current. The red circle, in lower left corner of the small green square marks the location of mooring CBeu. The green square represents the upstream 0.25 x 0.25 degrees area from which sea surface data have been obtained.

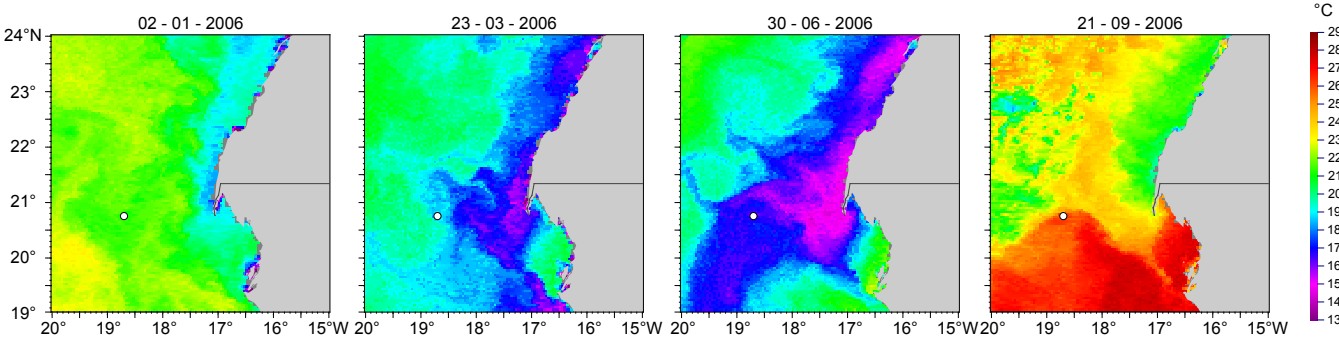

**Figure 2: Temperature distribution off Cap Blanc for four days, each from a different season. The location of sediment trap CBeu is indicated by a white circle. Colder waters near the African coast result from upwelling. The westward extension and location of colder upwelled waters is highly variable as is upwelling. Days of with strong (30-6-2006) or weak (02-01-2006) upwelling may occur any time of the year. From mid July to November a temperature pattern similar to that of 21-09-2006 is more typical. Data from ERDDAP, id: nceiPH53sstn1day_Lon0360.**

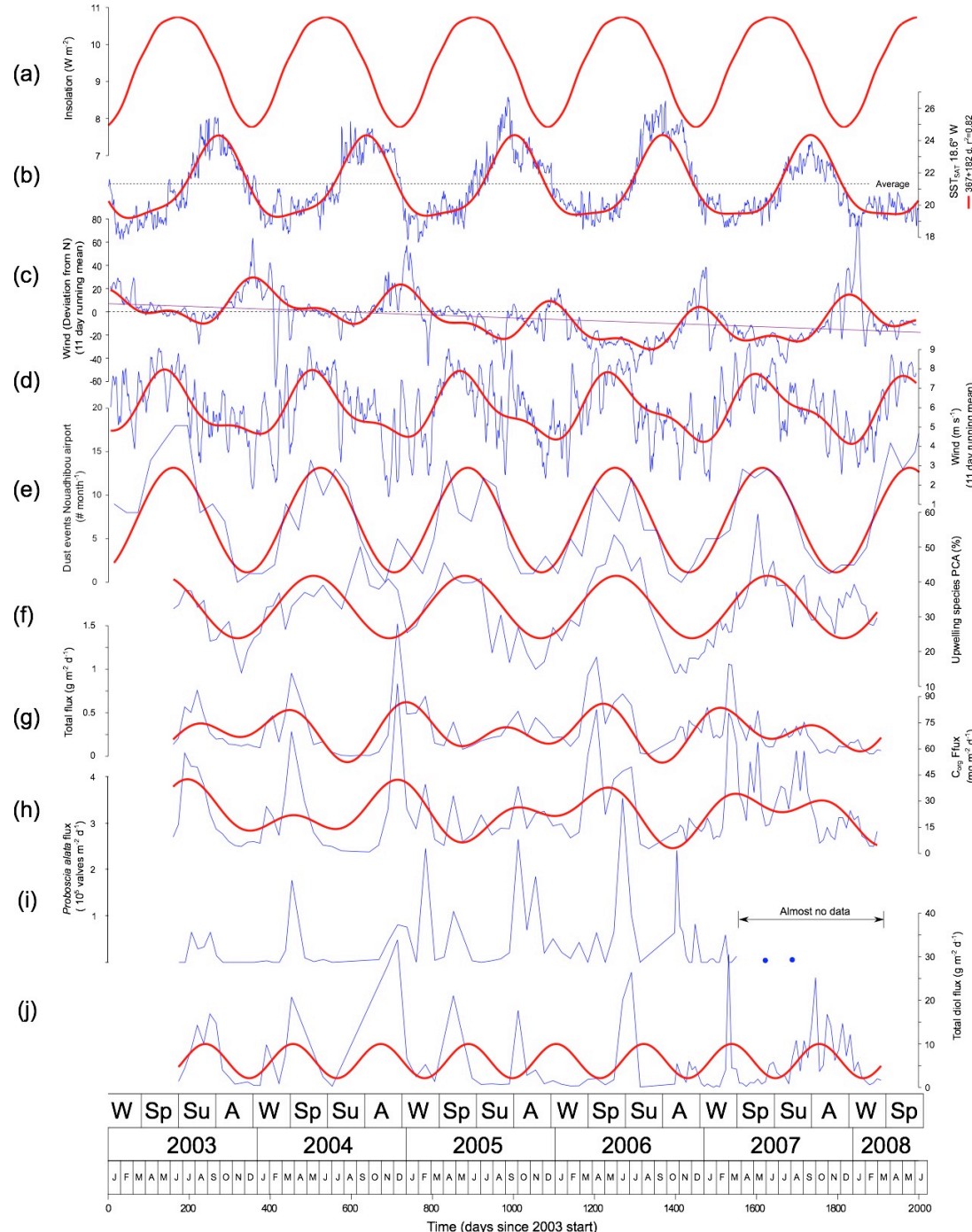


**Figure 3: Variation of external variables and the total diol flux through time for CBeu1–5. Thin blue lines, measured values (for wind parameters 11 days moving average); Thick red lines, the most important frequency components. Note that most parameters have a dominant annual cycle modulated by a semiannual cycle. The Total Diol Flux (graph J) is dominated by a 257 days cycle. For wind direction (c) the scale runs from east (positive), through north (zero) to west (negative).**





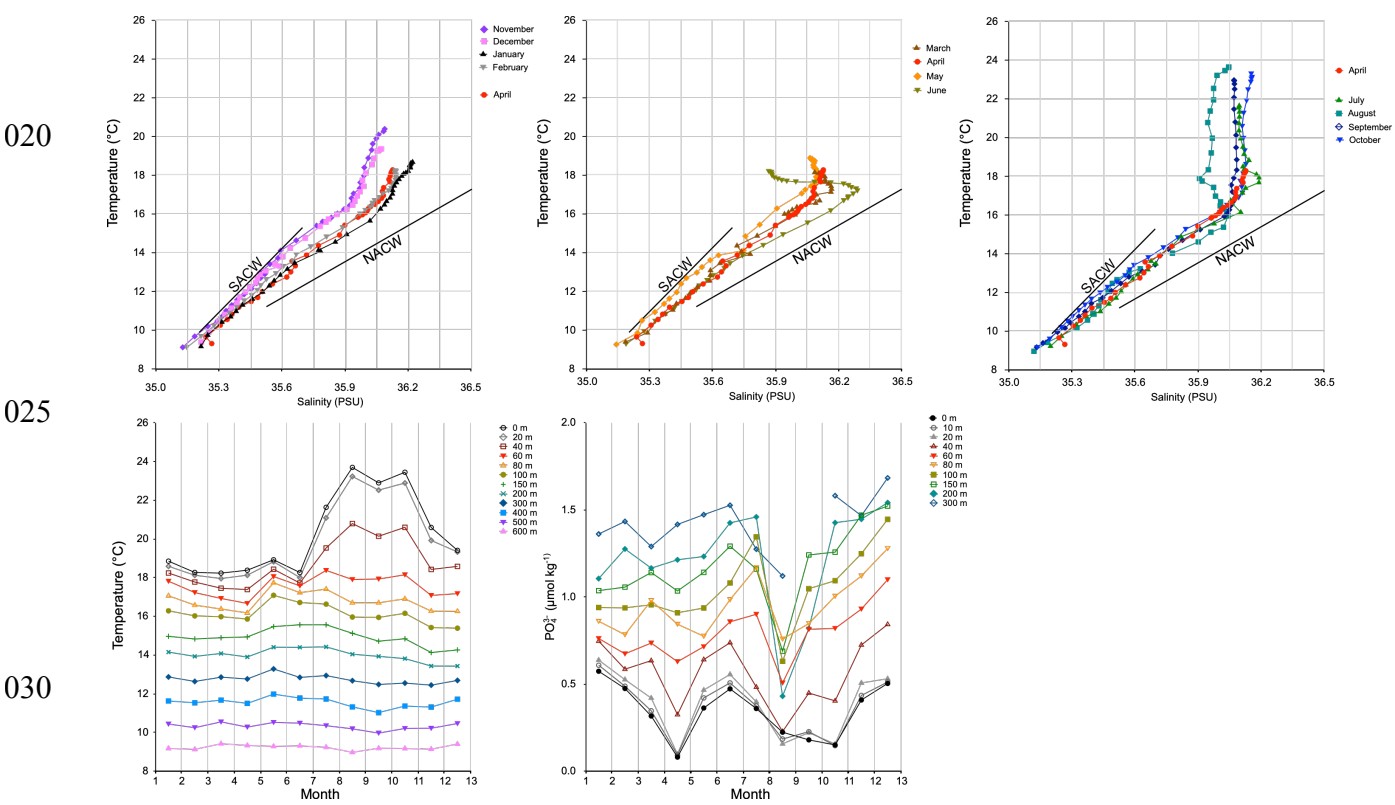


**Figure 4: Oceanographic data from the World Ocean Atlas 2018 (WOA). Upper three panels, Temperature-Salinity diagrams for each month for the 1° square centered at 20.5°N, 18.5°W. Solid lines represent the North Atlantic Central Water (NACW) and the South Atlantic Central Water (SACW) (Locarini et al., 2018). Lower left panel, water temperatures for different depths in the same 1° square (Zweng et al., 2019). Lower right panel, monthly $PO_4^{3-}$ concentrations against depth for the 5° square centered at 22.5°N, 17.5°W (Garcia et al., 2019).**

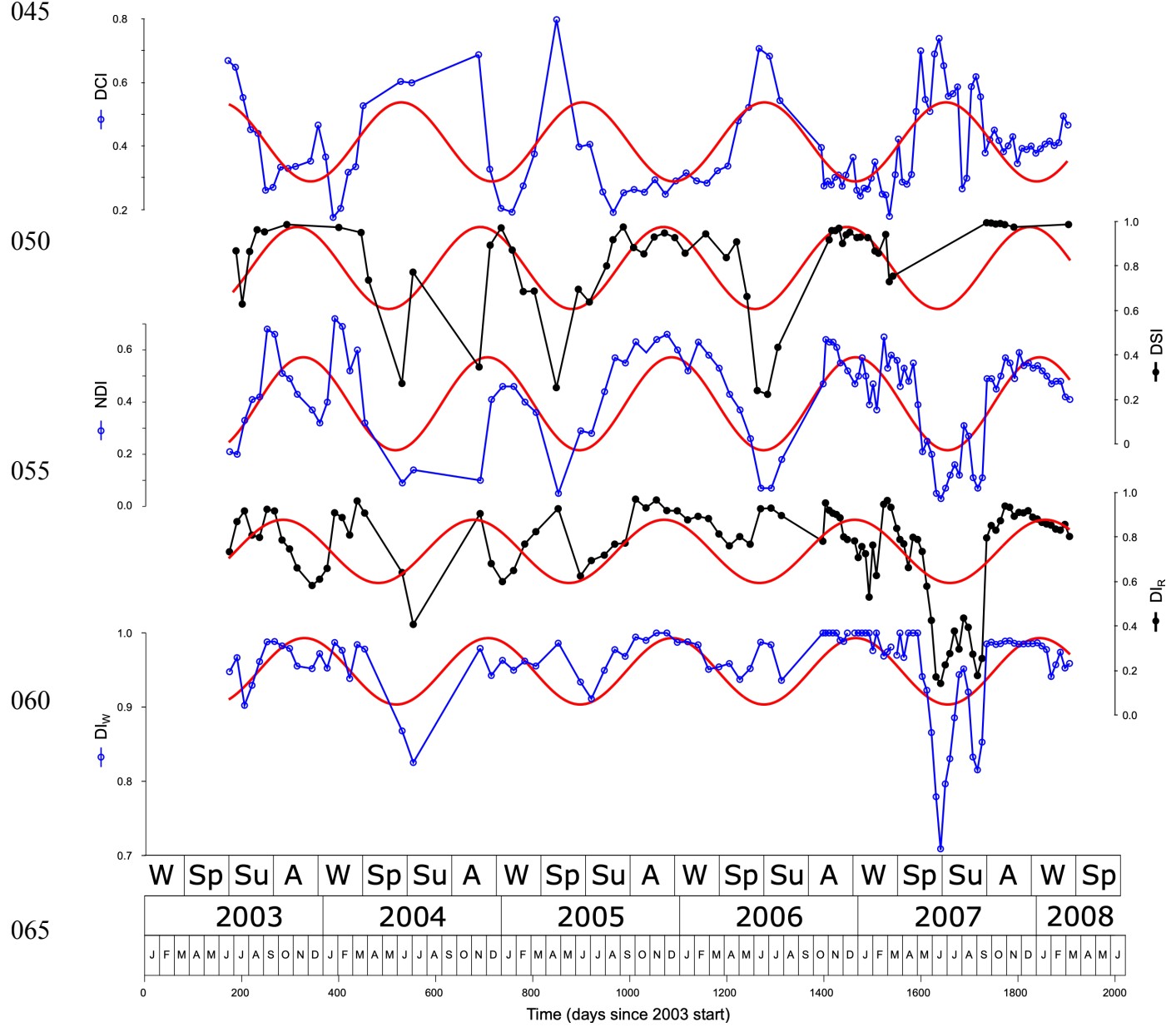

 **Figure 5: Diol proxies and their main (annual) frequency component. Note the close similarities between the diol proxies and the DCI behaving opposite to the others.**

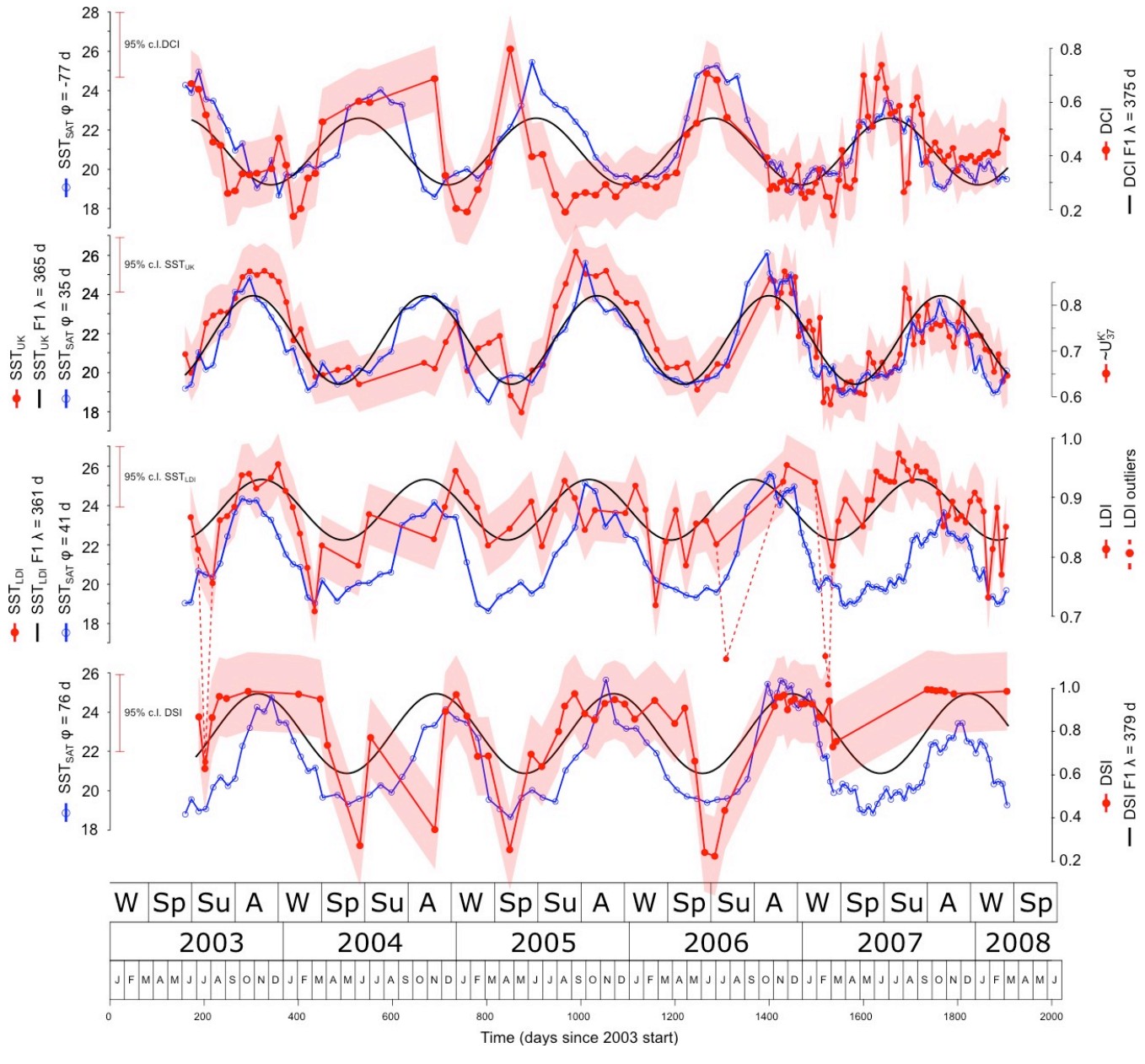

075

**Figure 6: Proxy-derived SST (red, filled circles), their annual frequency component (F1,, black) and satellite-derived SST$_{SAT}$ (blue, empty circles). The phase shifts of the SST$_{SAT}$ are such that correlation with the proxy derived SST is highest. Corresponding proxy values are indicated on the right side of the panels.  NB: check phase and λ LDI. Shaded red areas 95% confidence range.**

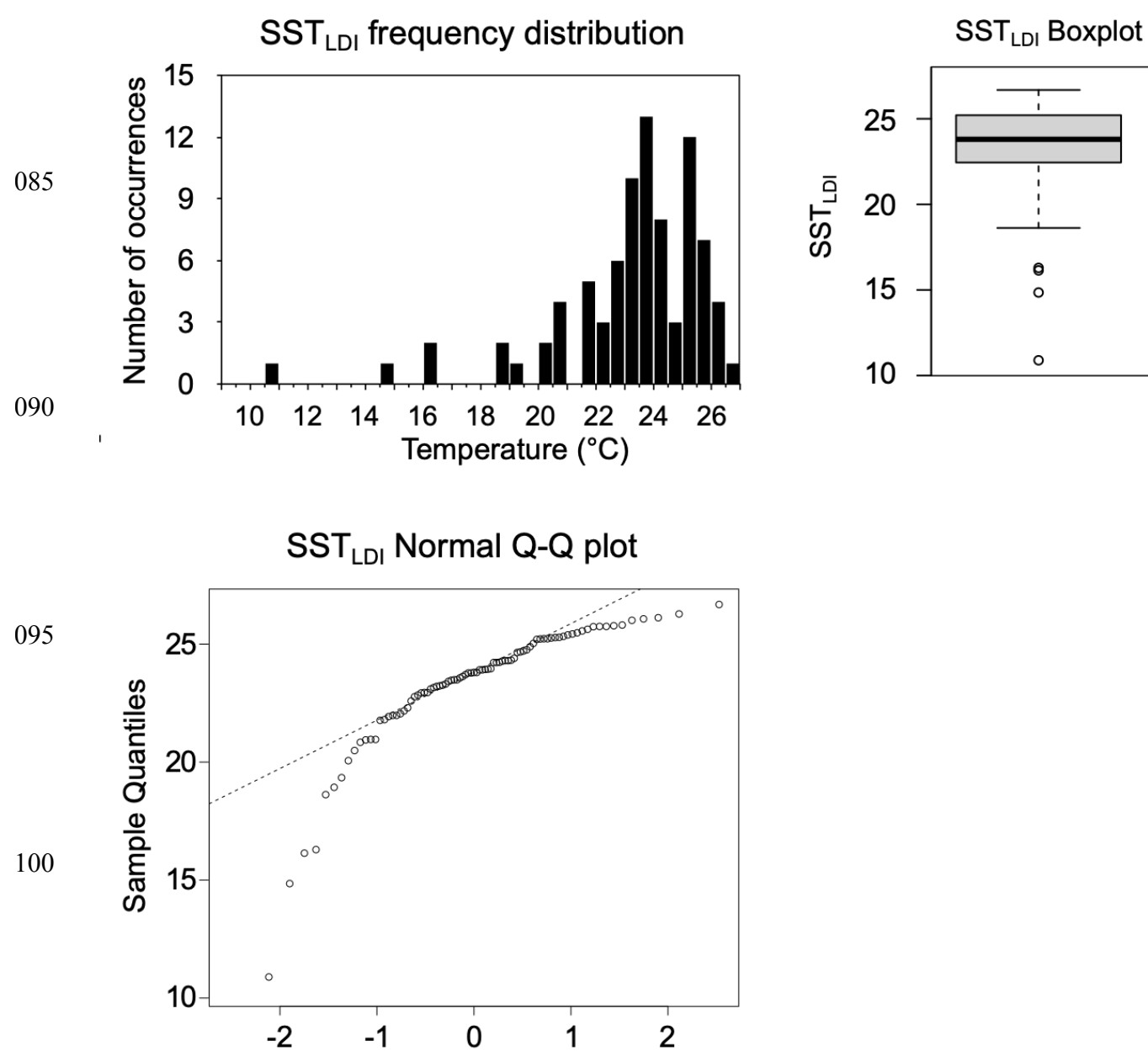

**Figure 7: Statistics of the SST$_{LDI}$ values identifying the 5 values below 16°C as outliers. A. Frequency distribution; B. Boxplot; C. Q–Q plot, with dotted line representing the expected values for a normally distributed dataset.**

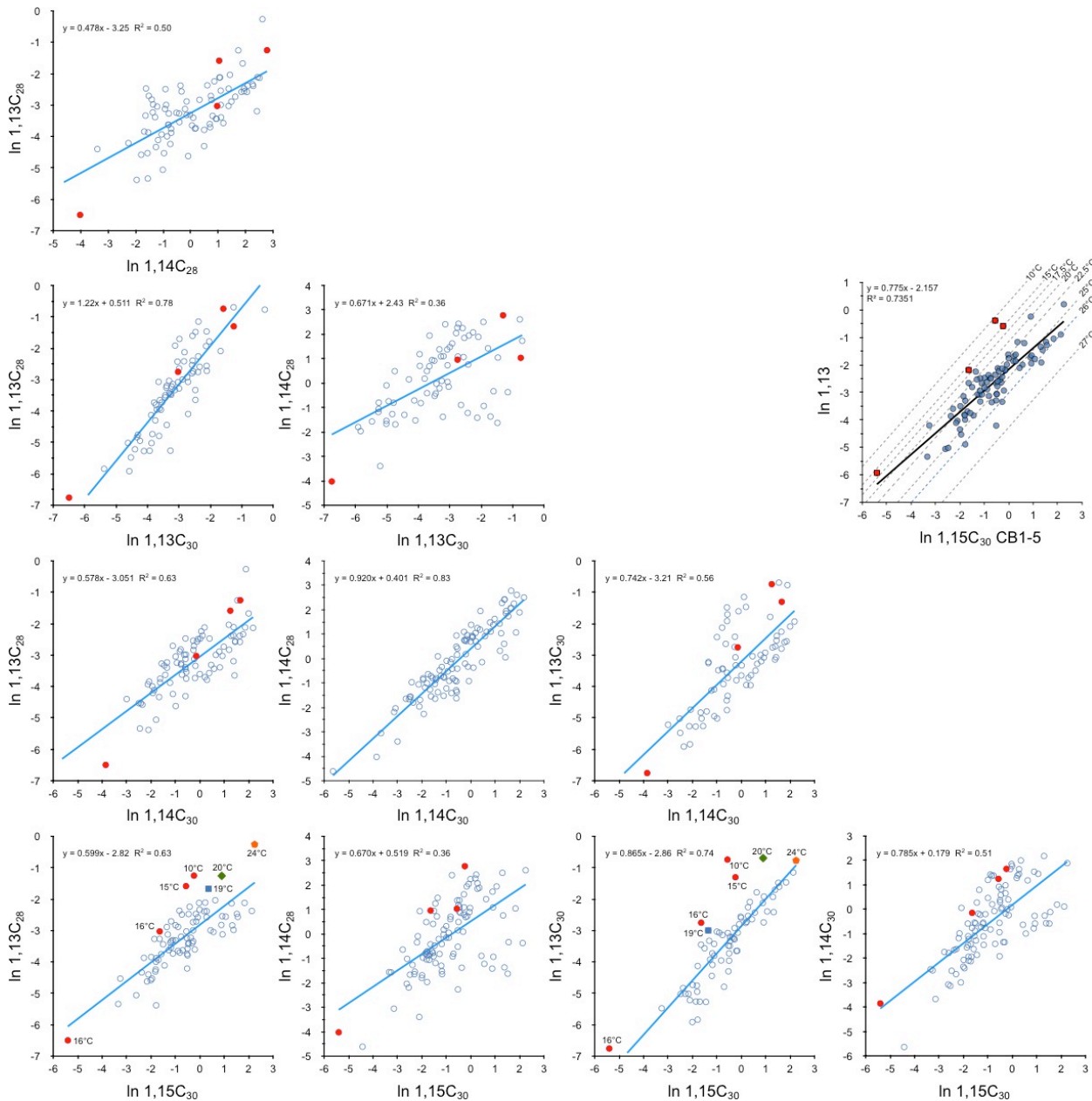

110

Figure 8: Cross plots of log transformed diol fluxes (µg m⁻² d⁻¹) with 1,13 diol values for samples providing SST$_{LDI}$ values below 18°C in solid red red. These values have not been included for obtaining the (blue) regression lines. In the 1,13 x 1,15 diol plots other extreme points are indicated by a blue square (19°C), green diamond (20°C) and orange pentagram (24°C). Upper right plot, the diols involved in the LDI (1,15C$_{30}$ and combined 1,13 diols) are provided. The dotted lines represent isotherms for the transfer
115 function of Rampen et al. (2012). The regression line shows how the (log transformed) composition of the diols involved in the LDI relates to temperature (excluding the outliers indicated in red).

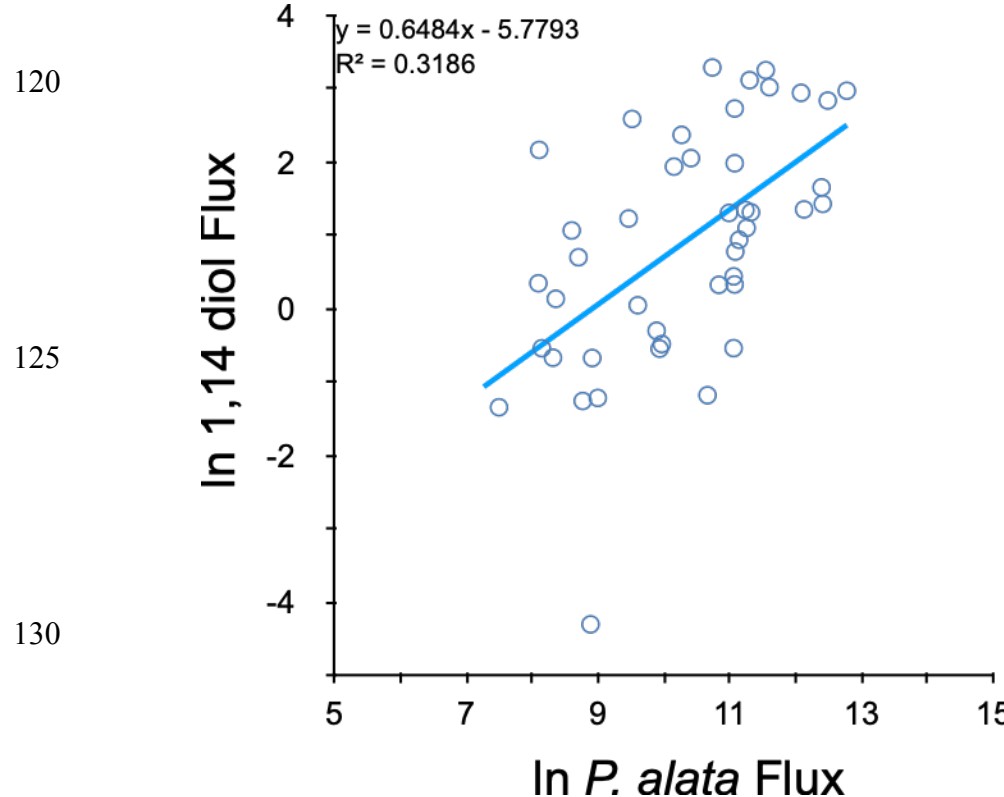

**Figure 9: Correlation of natural logarithms of 1,14 diols fluxes (µg m²d⁻) and diatom *Proboscia alata* fluxes (valves m² d⁻).**

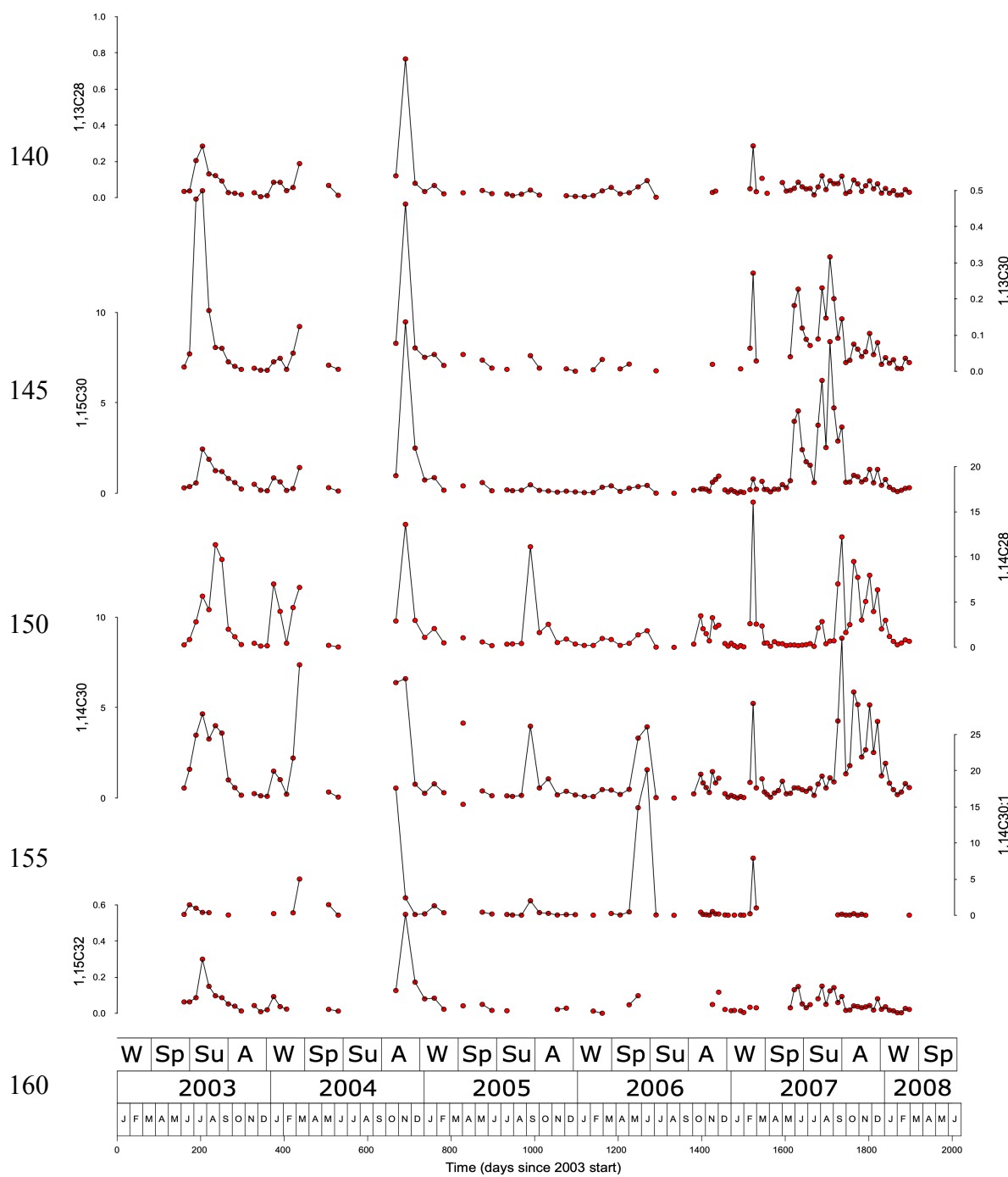

Figure 10: Fluxes for individual diols in µg m$^{-2}$ d$^{-1}$.

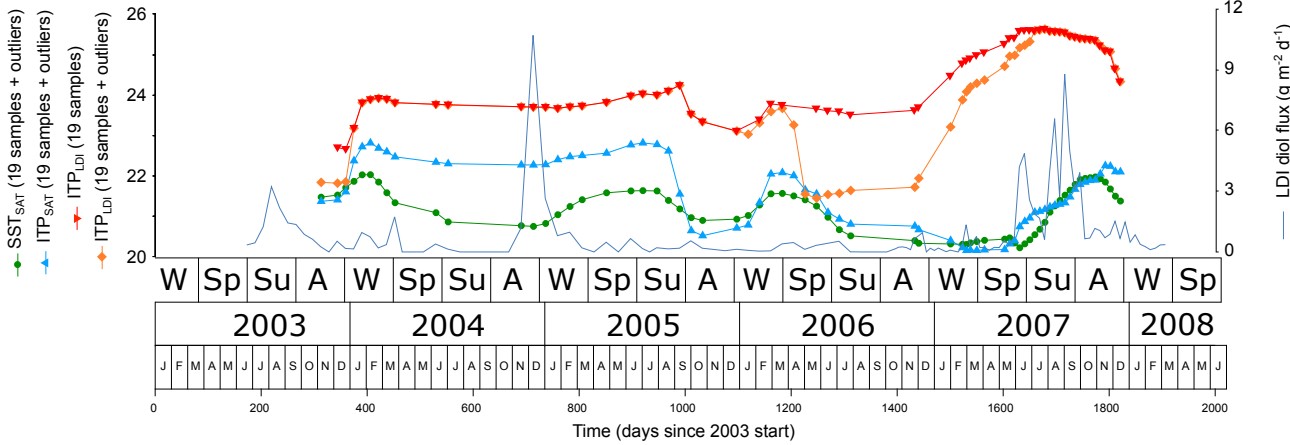

165

**Figure 11: Nineteen sample moving average SST$_{SAT}$ (green spheres) and nineteen sample moving integrated production temperatures for SST$_{SAT}$ (IPT$_{SAT}$, blue upward triangles) and for SST$_{LDI}$ (IPT$_{LDI}$, orange diamonds), both based on the LDI diol fluxes and the respective SST records. Samples with LDI outliers (values < 17°C) are included except for the upper graph of IPT$_{LDI}$ (red downward pointing triangles). For reference, the LDI diol flux is included as well.**

Table 1. Deployment data of sediment trap CBeu

| Mooring | Coordinates | GeoB No. cruise | Trap depth (m) | Ocean Bottom depth (m) | Sample amount | Capture duration (sample no. = days) | Sampling interval |
|---|---|---|---|---|---|---|---|
| 1 | 20°45'N 18°42'W | - POS 310 | 1296 | 2714 | 20 | 1=10, 2–20=15.5 | 5 Jun 2003– 5 Apr 2004 |
| 2 | 20°45'N 18°42'W | 9630-2 M 65-2 | 1296 | 2714 | 20 | 1–20=22 2–19=23 | 18 Apr 2004 – 20 Jul 2005 |
| 3 | 20°45.5'N 18°41.9'W | 11404-3 POS 344-1 | 1277 | 2693 | 20 | 21.5 | 25 Jul 2005 – 28 Sept 2006 |
| 4 | 20°45.7'N 18°42.4'W | 11835-2 MSM 04b | 1256 | 2705 | 20 | 1=3.5 2–20=7.5 | 28 Oct 2006 – 23 Mar 2007 |
| 5 | 20°44.9'N 18°42.7'W | 12910-2 POS 365-2 | 1263 | 2709 | 38 | 1,2=6.5 3–38=9.5 | 28 Mar 2007 – 17 Mar 2008 |

Table 2. Composition and position of the diols in the indices.

| Index \ Diol | $1,13C_{28}$ | $1,13C_{30}$ | $1,15C_{30}$ | $1,14C_{28}$ | $1,14C_{30}$ | $1,14C_{28:1}$[1] | $1,14C_{30:1}$ |
|---|---|---|---|---|---|---|---|
| LDI | D | D | N | | | | |
| DSI | | | | N | N | D | D |
| DCI | | | | D | N | | |
| DI$_R$ | | | D | N | N | | |
| DI$_W$ | D | D | | N | N | | |
| CDI | D | D | D | N | N | | |
| NDI | D | D | D | N | D | N | D |

Since all proxies are indices, all diols in the numerator (N) are also in the denominator (D).
[1]The $1,14C_{28:1}$ diol has not been detected in the CBeu1−5 samples

Table 3: Correlations of diol concentrations and their natural logarithms for CBeu1–5.

| | Lipid | $1,13C_{28}$ | $1,14C_{28}$ | $1,13C_{30}$ | $1,14C_{30:1}$ | $1,14C_{30}$ | $1,15C_{30}$ | $1,15C_{32}$ |
|---|---|---|---|---|---|---|---|---|
| | | | | linear correlations[1] ($r^2$) | | | | |
| | $1,13C_{28}$ | — | *0.35* | *0.59* | *0.01* | *0.31* | *0.44* | *0.58* |
| | $1,14C_{28}$ | 0.42 | — | *0.14* | *0.01* | *0.69* | *0.11* | *0.20* |
| | $1,13C_{30}$ | **0.75** | 0.29 | — | *0.01* | *0.19* | *0.70* | ***0.81*** |
| natural | $1,14C_{30:1}$ | 0.18 | 0.17 | 0.14 | — | *0.20* | *0.00* | *0.01* |
| logarithm | $1,14C_{30}$ | 0.58 | **0.83** | 0.50 | 0.33 | — | *0.12* | *0.20* |
| correlati- | $1,15C_{30}$ | 0.62 | 0.36 | **0.82** | 0.14 | 0.51 | — | ***0.84*** |
| ons[1] ($r^2$) | $1,15C_{32}$ | 0.70 | 0.34 | **0.84** | 0.17 | 0.51 | **0.85** | — |
| | 1,13 vs 1,14 | 0.48 | | | | | | |
| | 1,13 vs 1,15 | **0.79** | | | | | | |
| | 1,14 vs 1,15 | 0.45 | | | | | | |

Outliers: $1,13C_{28}$ and $1,13C_{30}$ of CBeu1-4, CBeu3-18, CBeu4-18 and CBeu4-19 excluded. Bold typeface, correlations >0.74.

[1]Linear correlations in italics.

Table 4: Correlations ($r^2$) between diol proxies and environment

| | $DI_W$ | $DI_R$ | CDI | NDI | DCI | DSI | LDI |
|---|---|---|---|---|---|---|---|
| $DI_R$ | **0.65** | | | | | | |
| CDI | **0.69** | **1.00** | | | | | |
| NDI | **0.49** | **0.46** | **0.48** | | | | |
| DCI | *0.32* | *0.15* | *0.16* | *0.74* | | | |
| DSI | 0.080 | *0.0004* | 0.0005 | **0.67** | *0.46* | | |
| LDI | *0.036* | *0.27* | *0.24* | *0.029* | *0.0003* | **0.09** | |
| ln *P. alata* Flux | **0.12** | **0.29** | **0.29** | 0.000 | 0.051 | *0.019* | |
| Total Flux | 0.011 | 0.024 | 0.021 | 0.001 | *0.017* | *0.013* | |
| % Upw spp. | *0.20* | *0.07* | **0.09** | *0.38* | **0.42** | *0.39* | *0.02* |
| Wind Speed | *0.19 φ 79* | *0.11 φ 79* | **0.093 φ 35** | *0.28 φ 32* | **0.29 φ 30** | *0.24 φ 21* | |
| SST (lead) | *0.23 φ -69* | *0.10 φ -69* | *0.13 φ -69* | *0.38 φ -77* | **0.29 φ -77** | *0.42 φ -124* | **(0.21 φ 140?)** |
| SST (lag) | **0.27 φ 94** | **0.13 φ 105** | **0.15 φ 104** | **0.34 φ 102** | *0.35 φ 117* | **0.25 φ 76** | **0.17 φ 41** |

Correlations with p<0.05 in bold, negative relations in italics. LDI outliers not included

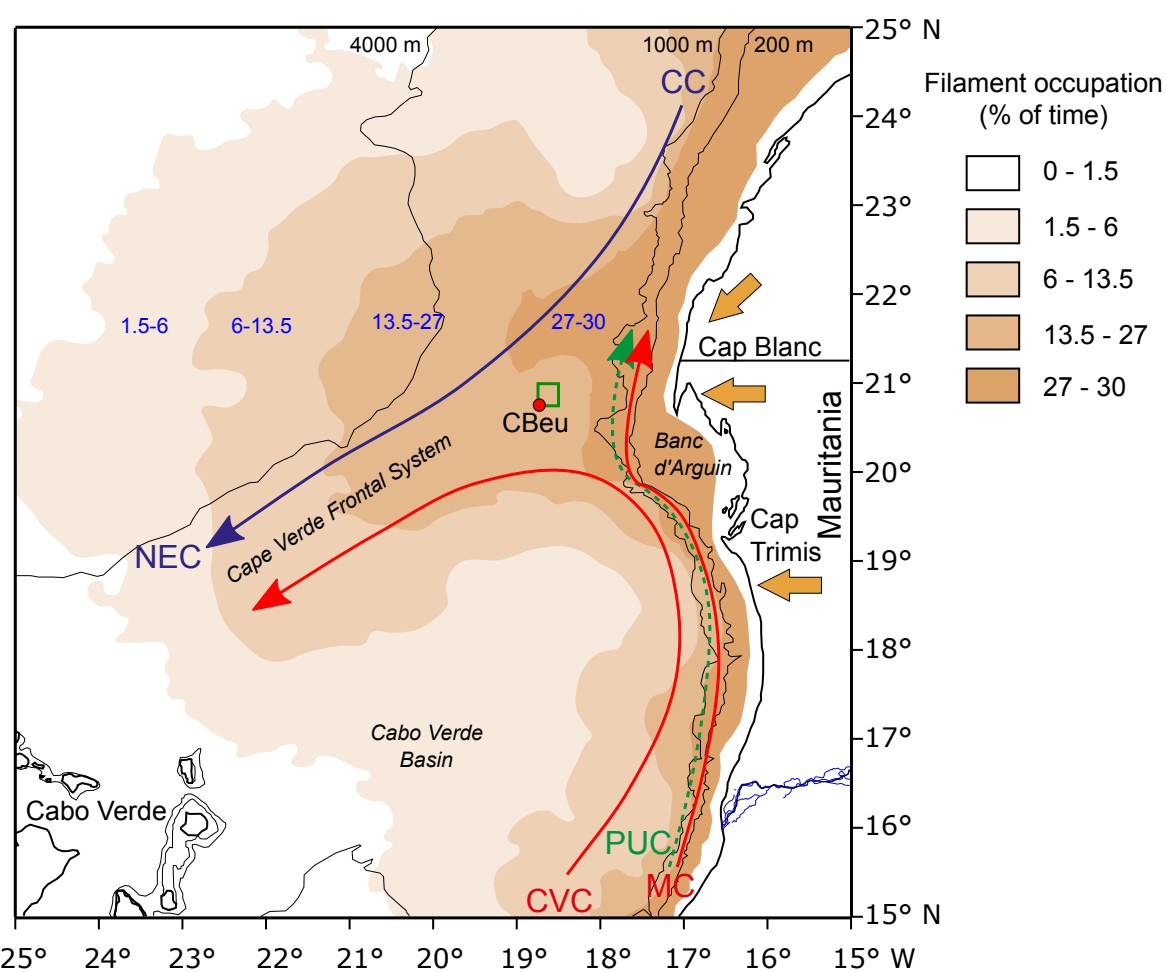

**Figure 1: The Mauritanian upwelling system. CC, Canary Current; NEC, North Equatorial Current; CVC, Cap Verde Current; MC, Mauritanian Current; PUC, Poleward Under Current. The red circle, in lower left corner of the small green square marks the location of mooring CBeu. The green square represents the upstream 0.25 x 0.25 degrees area from which sea surface data have been obtained.**

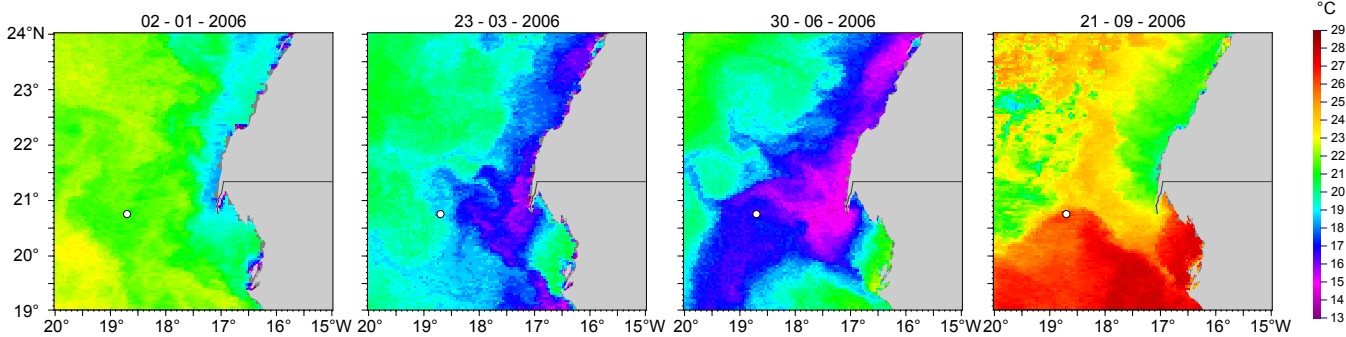

**Figure 2: Temperature distribution off Cap Blanc for four days, each from a different season. The location of sediment trap CBeu is indicated by a white circle. Colder waters near the African coast result from upwelling. The westward extension and location of colder upwelled waters is highly variable as is upwelling. Days of with strong (30-6-2006) or weak (02-01-2006) upwelling may occur any time of the year. From mid July to November a temperature pattern similar to that of 21-09-2006 is more typical. Data from ERDDAP, id: nceiPH53sstn1day_Lon0360.**

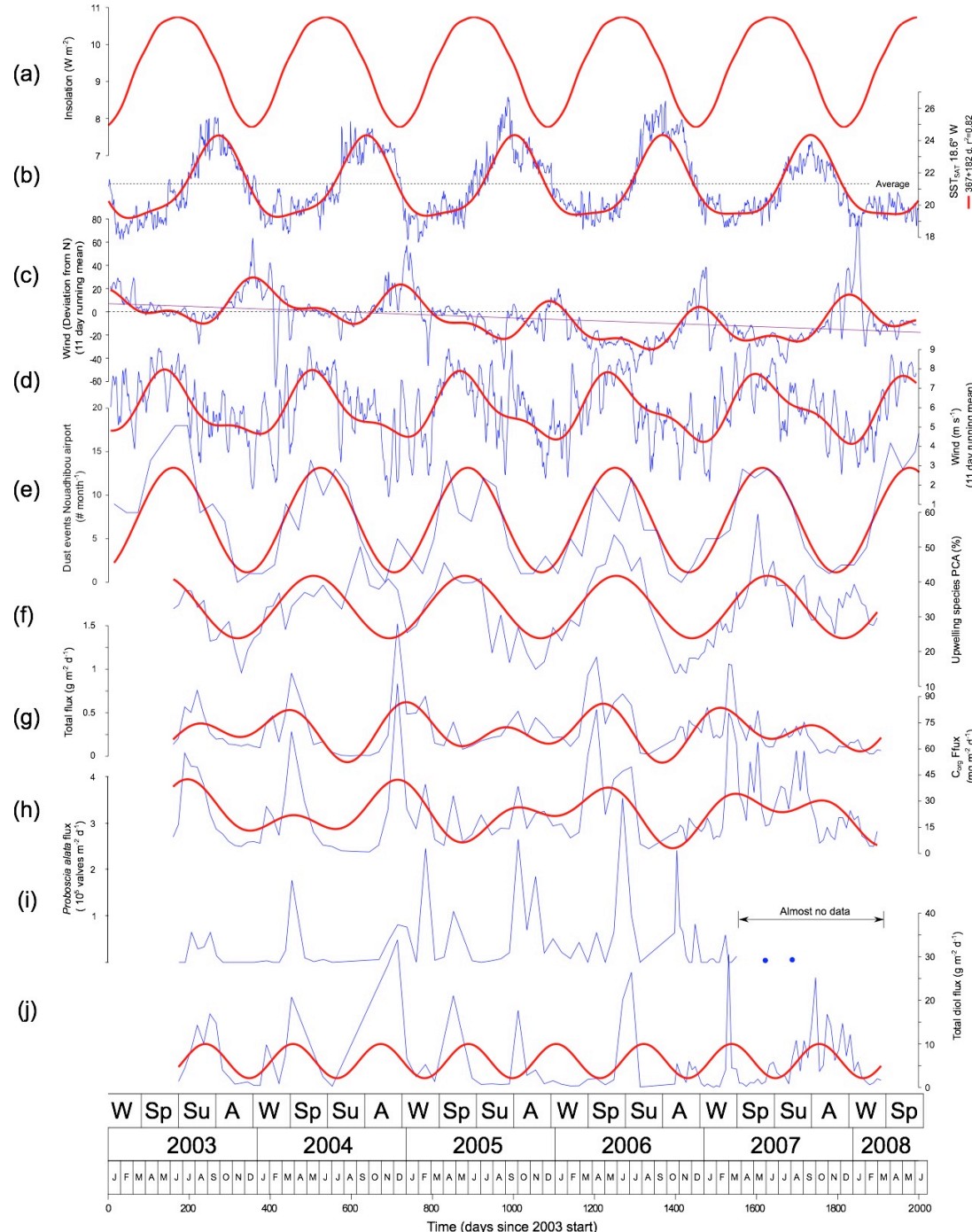

**Figure 3: Variation of external variables and the total diol flux through time for CBeu1–5.** Thin blue lines, measured values (for wind parameters 11 days moving average); Thick red lines, the most important frequency components. Note that most parameters have a dominant annual cycle modulated by a semiannual cycle. The Total Diol Flux (graph J) is dominated by a 257 days cycle. For wind direction (c) the scale runs from east (positive), through north (zero) to west (negative).

010

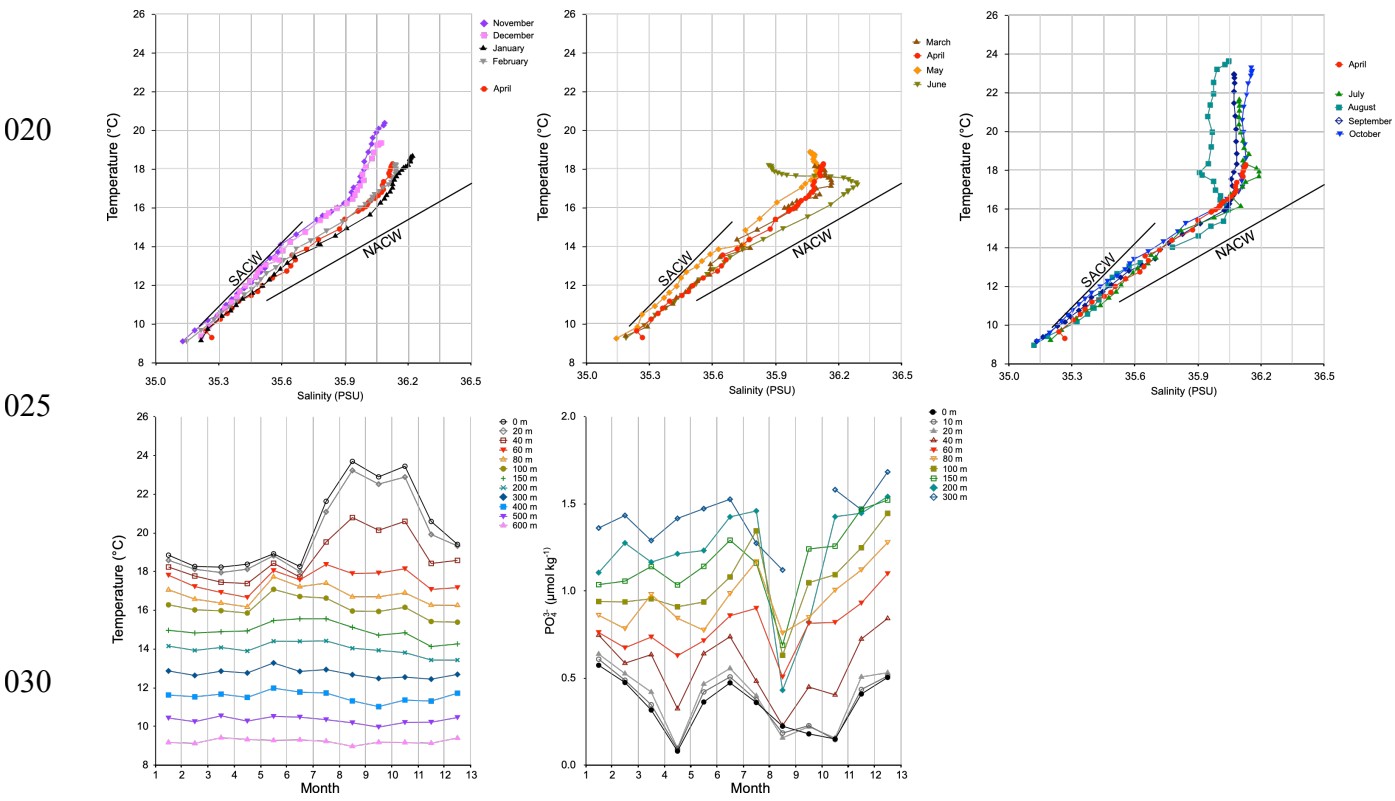

**Figure 4: Oceanographic data from the World Ocean Atlas 2018 (WOA). Upper three panels, Temperature-Salinity diagrams for each month for the 1° square centered at 20.5°N, 18.5°W. Solid lines represent the North Atlantic Central Water (NACW) and the South Atlantic Central Water (SACW) (Locarini et al., 2018). Lower left panel, water temperatures for different depths in the same 1° square (Zweng et al., 2019). Lower right panel, monthly $PO_4^{3-}$ concentrations against depth for the 5° square centered at 22.5°N, 17.5°W (Garcia et al., 2019).**

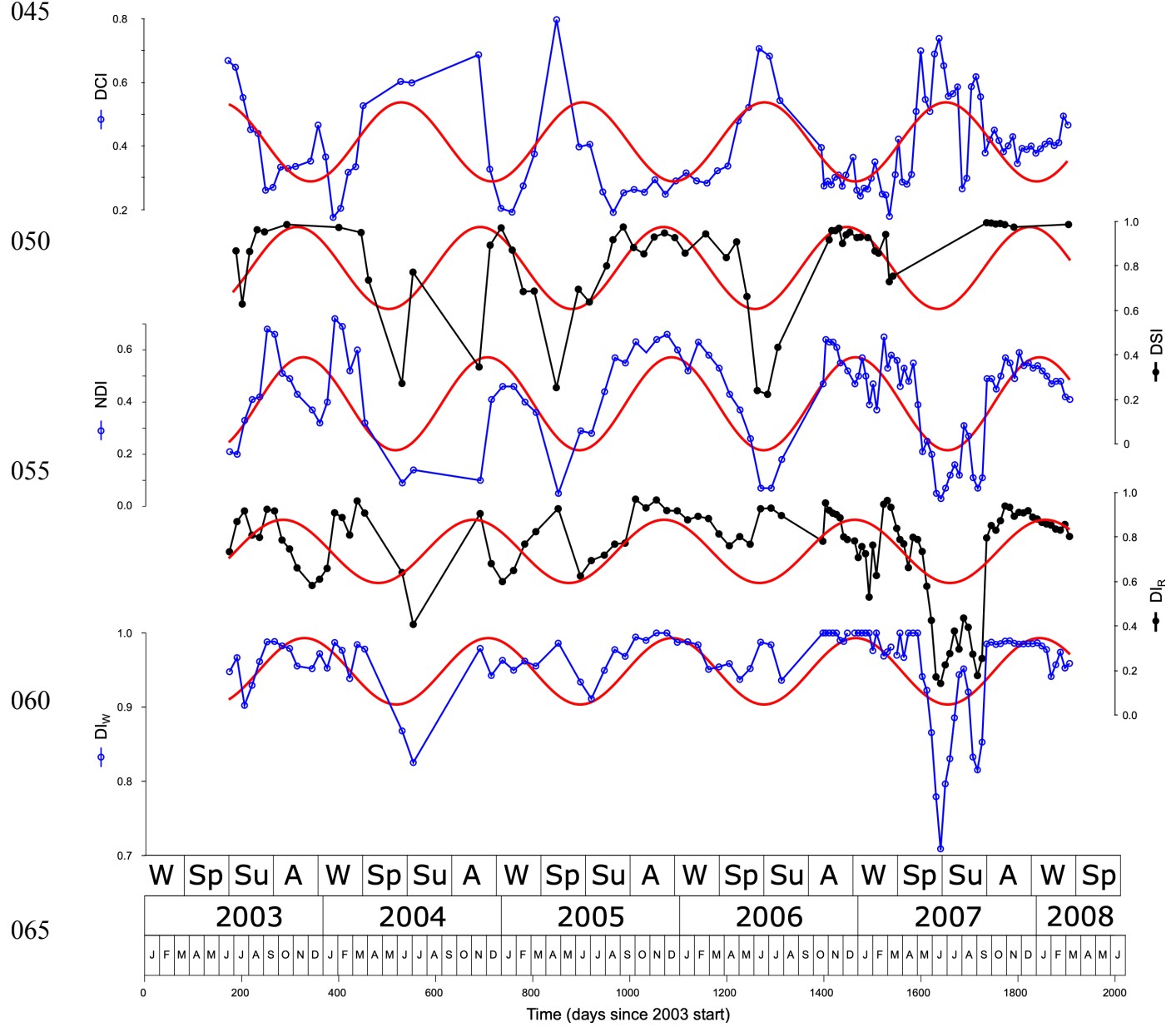

**Figure 5: Diol proxies and their main (annual) frequency component. Note the close similarities between the diol proxies and the DCI behaving opposite to the others.**

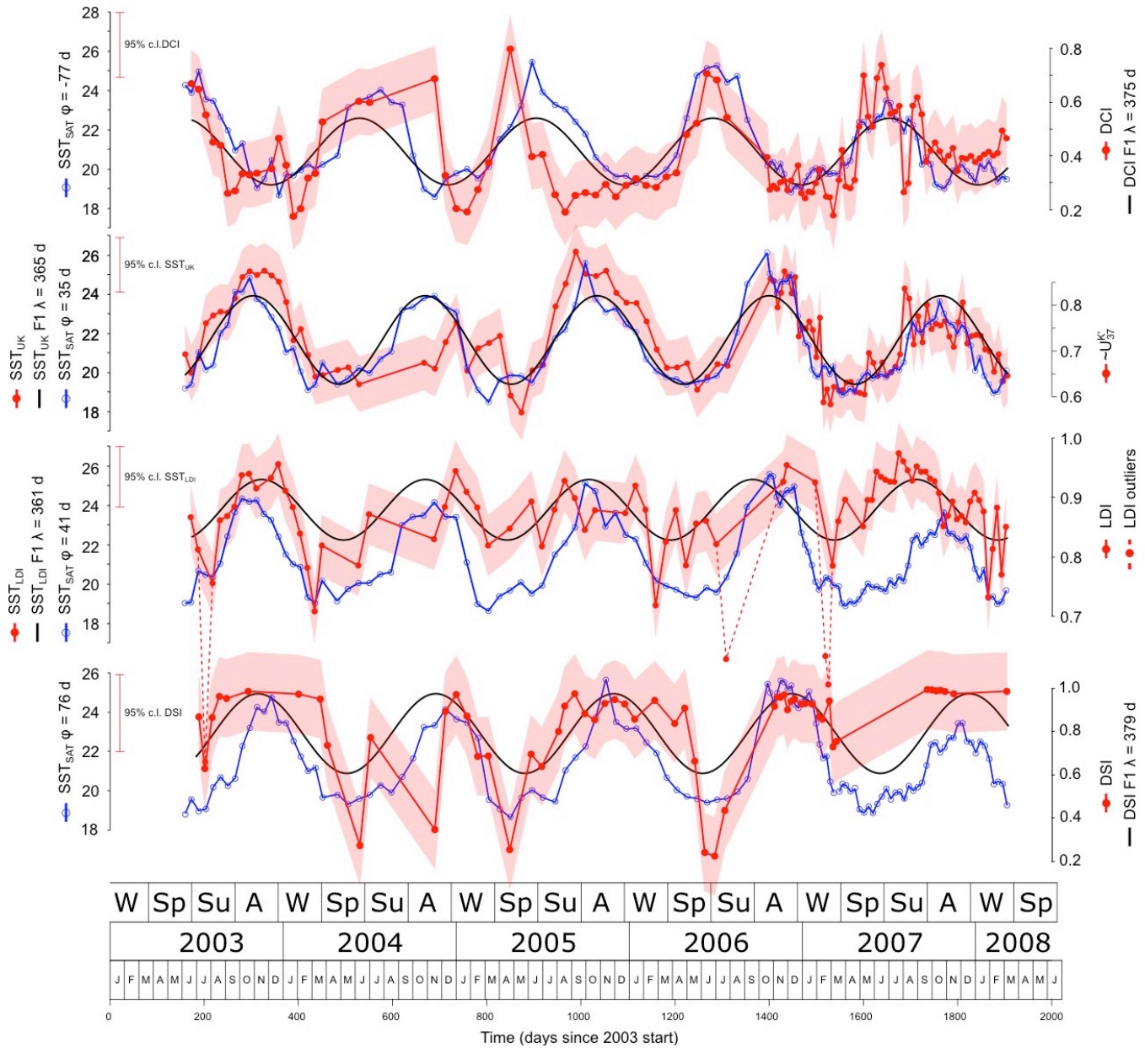


**Figure 6: Proxy-derived SST (red, filled circles), their annual frequency component (F1,, black) and satellite-derived SST$_{SAT}$ (blue, empty circles). The phase shifts of the SST$_{SAT}$ are such that correlation with the proxy derived SST is highest. Corresponding proxy values are indicated on the right side of the panels. NB: check phase and λ LDI. Shaded red areas 95% confidence range.**







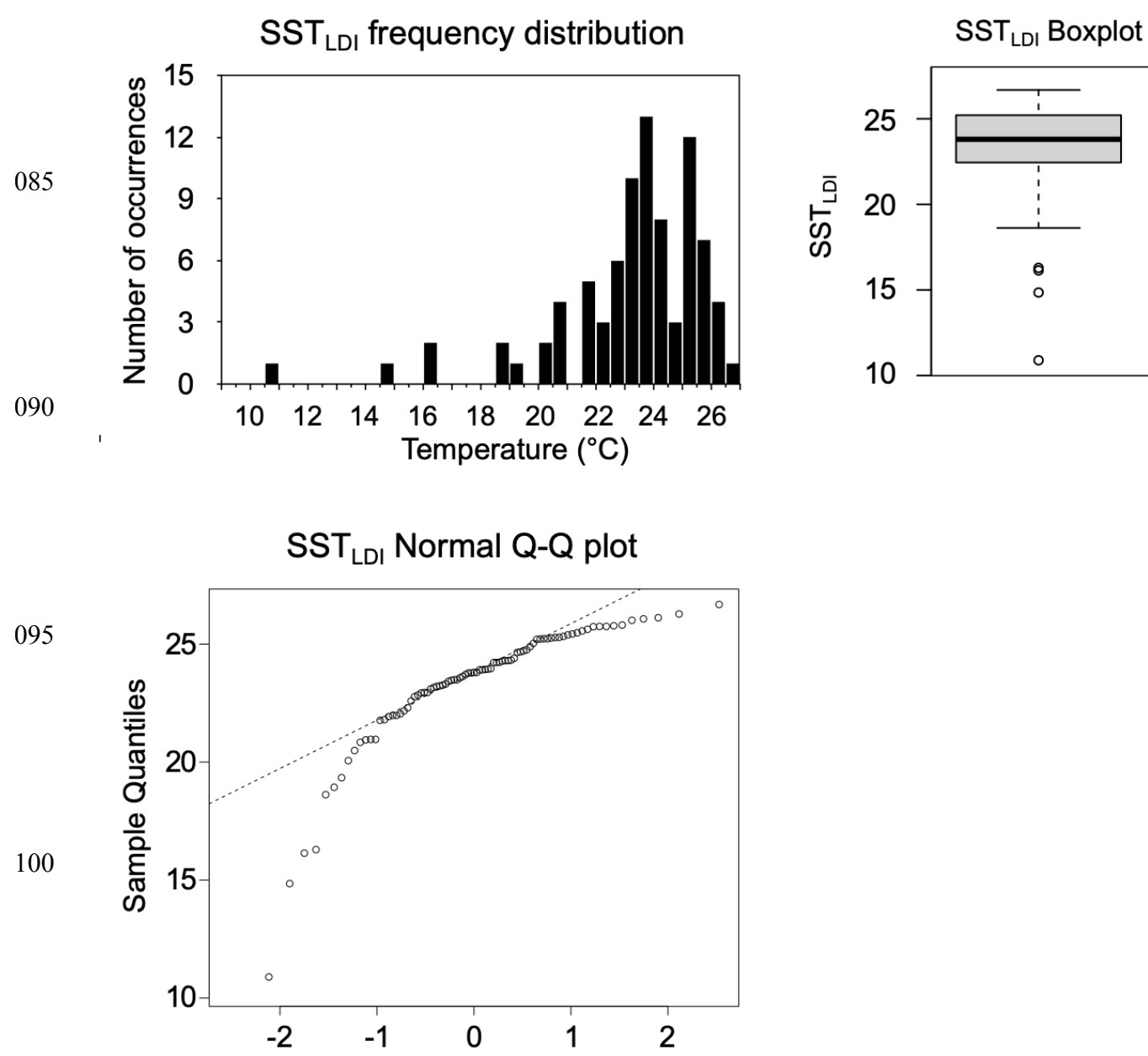

**Figure 7: Statistics of the SST$_{LDI}$ values identifying the 5 values below 16°C as outliers. A. Frequency distribution; B. Boxplot; C. Q–Q plot, with dotted line representing the expected values for a normally distributed dataset.**


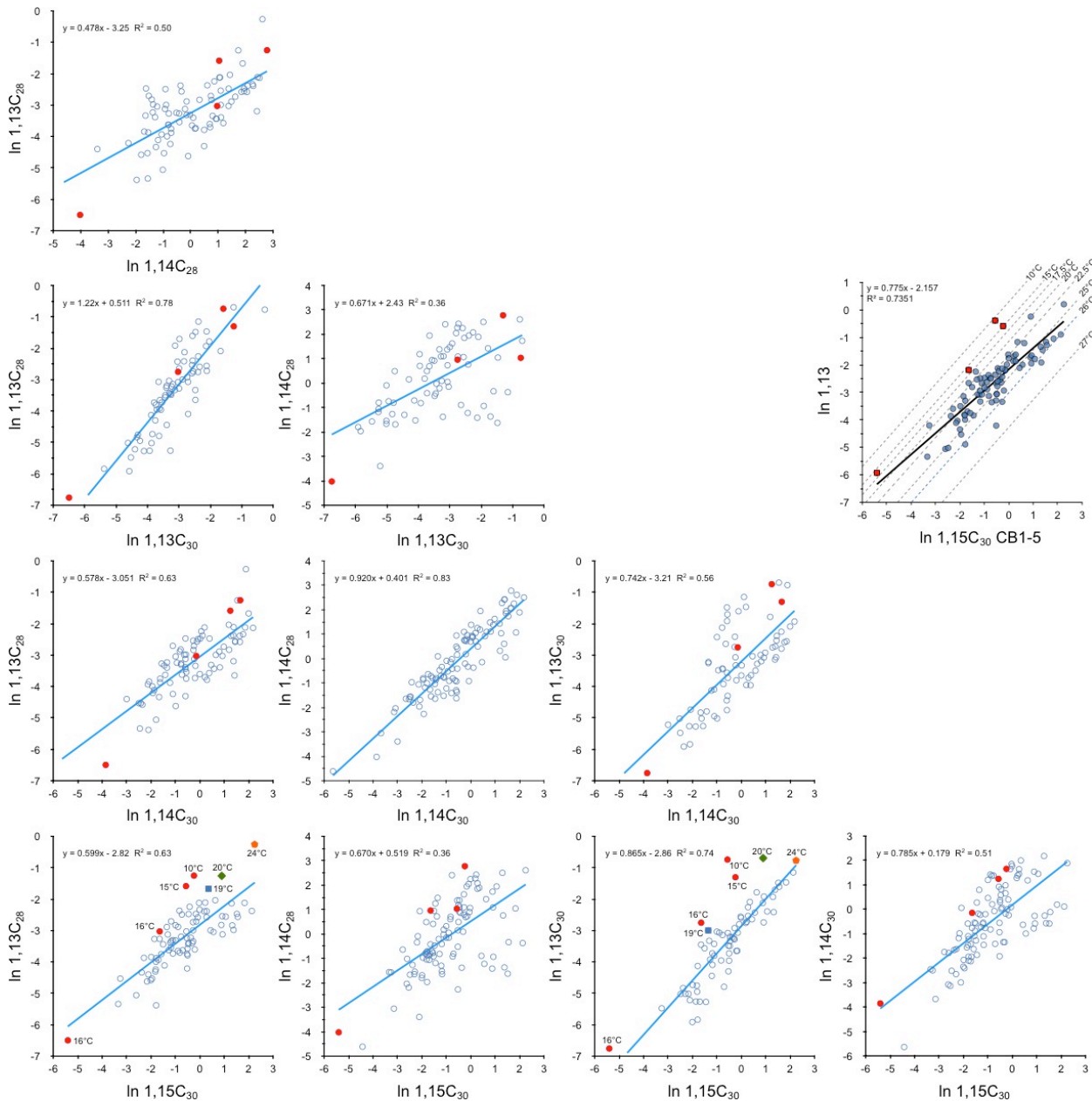

**Figure 8: Cross plots of log transformed diol fluxes (µg m⁻² d⁻¹) with 1,13 diol values for samples providing SST$_{LDI}$ values below 18°C in solid red red. These values have not been included for obtaining the (blue) regression lines. In the 1,13 x 1,15 diol plots other extreme points are indicated by a blue square (19°C), green diamond (20°C) and orange pentagram (24°C). Upper right plot, the diols involved in the LDI (1,15C$_{30}$ and combined 1,13 diols) are provided. The dotted lines represent isotherms for the transfer function of Rampen et al. (2012). The regression line shows how the (log transformed) composition of the diols involved in the LDI relates to temperature (excluding the outliers indicated in red).**


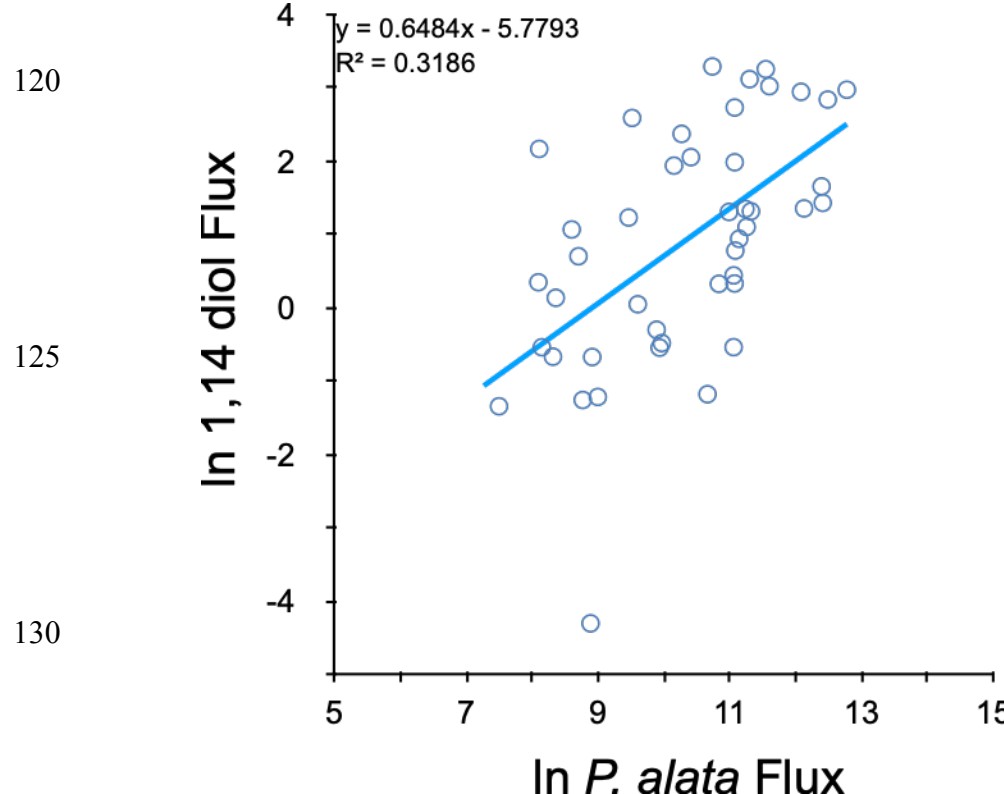

**Figure 9: Correlation of natural logarithms of 1,14 diols fluxes (µg m⁻²d⁻¹) and diatom *Proboscia alata* fluxes (valves m⁻² d⁻¹).**





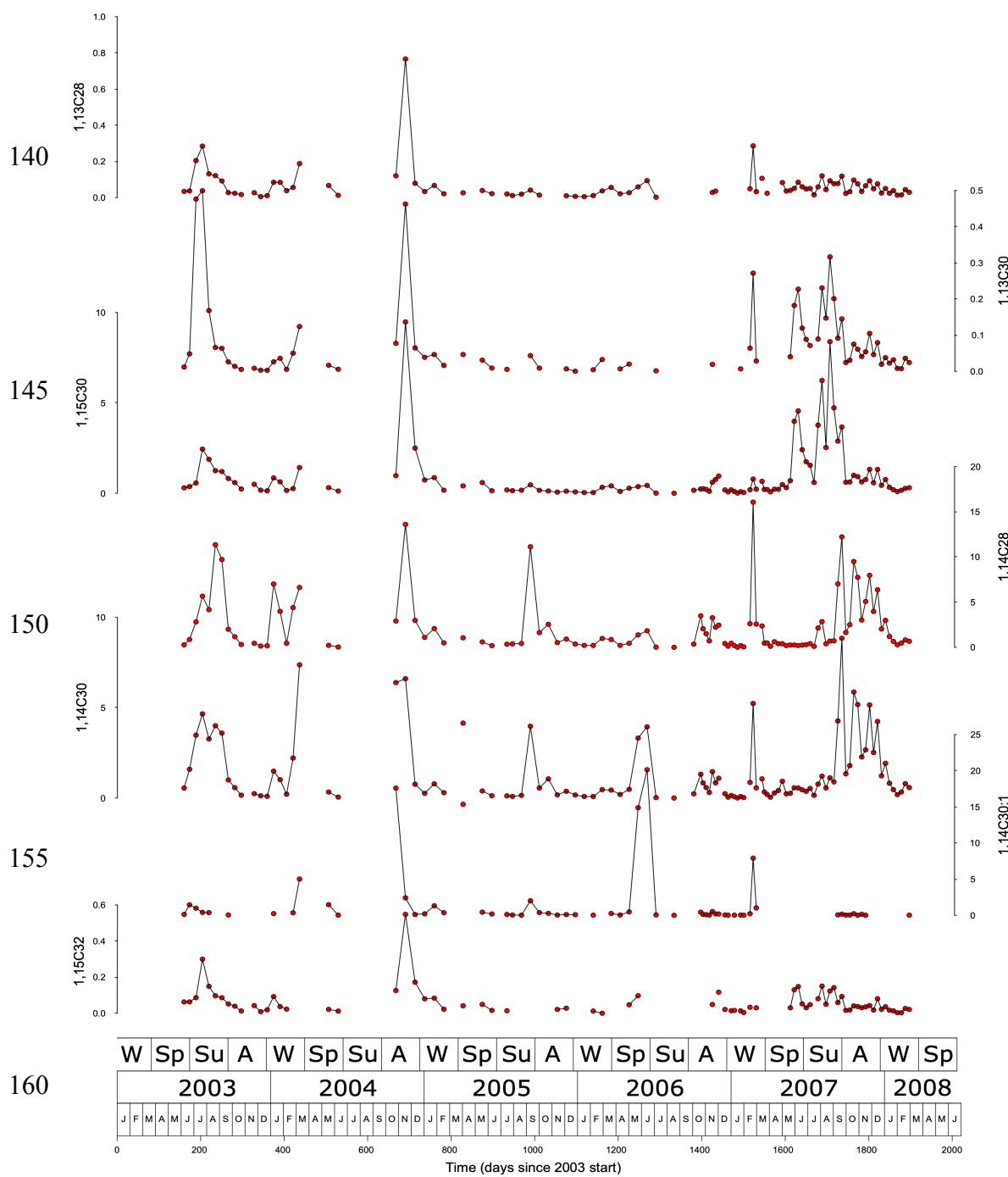

Figure 10: Fluxes for individual diols in µg m$^{-2}$ d$^{-1}$.

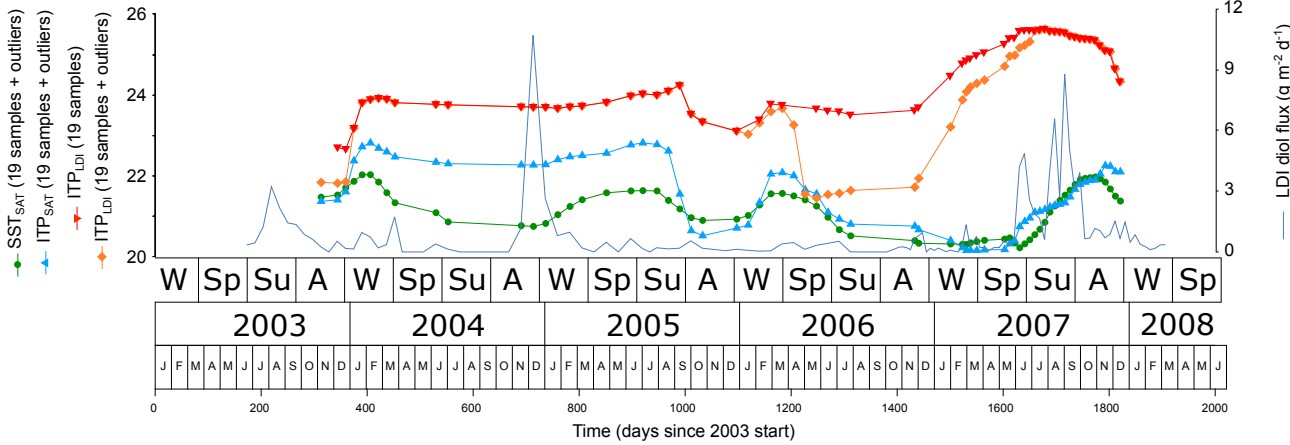

165

**Figure 11: Nineteen sample moving average SST$_{SAT}$ (green spheres) and nineteen sample moving integrated production temperatures for SST$_{SAT}$ (IPT$_{SAT}$, blue upward triangles) and for SST$_{LDI}$ (IPT$_{LDI}$, orange diamonds), both based on the LDI diol fluxes and the respective SST records. Samples with LDI outliers (values < 17°C) are included except for the upper graph of IPT$_{LDI}$ (red downward pointing triangles). For reference, the LDI diol flux is included as well.**