# Peer review of "Performance of long-chain alkane—1,mid—chain—diol based temperature and productivity proxies at test: a five-years sediment trap record from the Mauritanian upwelling"

_Biogeosciences, 2021_

## Author Comment (AC1)

[Figure]

**Figure RC2-1: Temperature distribution off Cap Blanc for four days, each from a different season. The location of sediment trap CBeu is indicated by a white circle. Colder waters near the African coast result from upwelling. The westward extension and location of colder upwelled waters is highly variable as is upwelling. Days of with strong (30-6-2006) or weak (02-01-2006) upwelling may occur any time of the year. From mid July to November a temperature pattern similar to that of 21-09-2006 is more typical.**

---

## Author Comment (AC2)

[Figure]

**Figure RC1-1. Nineteen sample moving average SST$_{SAT}$ (green spheres) and nineteen sample moving integrated production temperatures for SST$_{SAT}$ (IPT$_{SAT}$, blue upward triangles) and for SST$_{LDI}$ (IPT$_{LDI}$, orange diamonds), both based on the LDI diol fluxes and the respective SST records. Samples with LDI outliers (values < 17°C) are included except for the upper graph of IPT$_{LDI}$ (red downward pointing triangles). For reference, the LDI dot flux is included as well.**

---

## Author Response (AR1)

**Authors Response bg-2021-309**

**Dear Editor,**

We changed the manuscript according to the recommendations of the reviewers. The major changes are listed below. Minor changes made in response to the reviewers and other small corrections have been highlighted in the annotated manuscript. A detailed listing of all changes is provided in both reports submitted in response to the reviews and we presume we should not repeat that here.

1. In the title 'diols' has been changed in 'alkan-1,mid-chain' diols to better indicate the nature of the molecules and the positions of the hydroxy groups.

2. In the abstract several changes have been made in the second half. The link between the DSI proxy, upwelling and temperature has been highlighted and the observation that the LDI reflects summer temperatures in this setting. The order of the final statements has been changed so that now the last sentence focuses on diol proxies in stead of the alkenone-based UK'37.

3. Paragraph 2.6. The detail has been added on the kind of diatom data obtained.

4. Paragraph 2.7.2. Four daily maps of SST distribution, one for each season have been added (fig. 2) as recommended by the reviewers.

5. This addition has been further specified in the results section, paragraph 3.1.1.

6. Paragraph 3.4 has been largely rewritten

7. At the end of paragraph 3.4.1 we added an explanation for not including DSI values for samples where we could not detect unsaturated diols.

8. The first part of paragraph 4.2 was simplified as requested by the reviewers.

9. A new paragraph, 4.3 (matching 3.3) on diol fluxes was added as recommended by one of the reviewers.

10. In paragraph 4.4 we added a few sentences (lines 777-783) on the roles of upwelling and insolation on the regional temperatures and the hypothesis that the DSI is primarily reflecting the upwelling related temperatures and to a much lesser extent represents the more oligotrophic insolation dominated period of high summer SST.

11. We also added a paragraph 4.5 on the integrated production temperatures of the LDI and the effects of diol productivity changes and outliers on the SSTLDI if averaged over a period of about a year or longer as is illustrated in a new figure (Fig. 11). We did not consider this for the other diol proxies since this would largely be a repetition of arguments.

12. At the end of paragraph 4.6. a statement has been added hypothesizing that the LDI at the site of CBeu primarily reflects SST prevailing in the more oligotrophic waters which is summer in this case.

13. The conclusions have been adapted accordingly.

Figure 6. error bars have been added.

In the supplementary information a two figures have been added. S4 demonstrates logarithmic nature of the flux amplitude of LDI relevant diols, with the trap cups in order of decreasing flux.
S5 demonstrates the change in LDI upon adding individual trap cups with successively less (or more) diols starting with the cup with the highest (lowest) flux.

Sincerely,

Gerard Versteegh